# Primary and promiscuous functions coexist during evolutionary innovation through whole protein domain acquisitions

**José Antonio Escudero[1,2,3,4], Aleksandra Nivina[1,2,5], Harry E Kemble[6], Céline Loot[1,2], Olivier Tenaillon[6], Didier Mazel[1,2]***

[1]Institut Pasteur, Unité de Plasticité du Génome Bactérien, Département Génomes et Génétique, Paris, France; [2]CNRS, UMR3525, Paris, France; [3]Molecular Basis of Adaptation, Departamento de Sanidad Animal, Facultad de Veterinaria, Universidad Complutense de Madrid, Madrid, Spain; [4]VISAVET Health Surveillance Centre. Universidad Complutense Madrid. Avenida Puerta de Hierro, Madrid, Spain; [5]Université Paris Descartes, Sorbonne Paris Cité, Paris, France; [6]Infection, Antimicrobials, Modelling, Evolution, INSERM, UMR 1137, Université Paris Diderot, Université Paris Nord, Paris, France

**Abstract** Molecular examples of evolutionary innovation are scarce and generally involve point mutations. Innovation can occur through larger rearrangements, but here experimental data is extremely limited. Integron integrases innovated from double-strand- toward single-strand-DNA recombination through the acquisition of the I2 α-helix. To investigate how this transition was possible, we have evolved integrase IntI1 to what should correspond to an early innovation state by selecting for its ancestral activity. Using synonymous alleles to enlarge sequence space exploration, we have retrieved 13 mutations affecting both I2 and the multimerization domains of IntI1. We circumvented epistasis constraints among them using a combinatorial library that revealed their individual and collective fitness effects. We obtained up to $10^4$-fold increases in ancestral activity with various asymmetrical trade-offs in single-strand-DNA recombination. We show that high levels of primary and promiscuous functions could have initially coexisted following I2 acquisition, paving the way for a gradual evolution toward innovation.

**\*For correspondence:**
mazel@pasteur.fr

**Competing interests:** The authors declare that no competing interests exist.

## Introduction

Integrons are genetic elements that enhance bacterial evolvability through the acquisition of new genes encoded in integron *cassettes* (*Francia et al., 1997*; *Escudero et al., 2015*; *Mazel, 2006*; *Hall and Stokes, 1993*; *Figure 1a*). Integrons can stockpile arrays of usually promoter-less cassettes (*Loot et al., 2017*), and modulate their expression through shuffling events, providing a low-cost adaptive memory for the cell (*Lacotte et al., 2017*). Although originally located on chromosomes (*Mazel et al., 1998*; *Rowe-Magnus et al., 2001*), some integrons have been mobilized onto conjugative plasmids and have reached our hospitals (*Escudero and Mazel, 2017*; *Rowe-Magnus et al., 2003*; *Ghaly et al., 2017*) where they currently play a major role in the ecology of multidrug resistance among Gram negative pathogens (*Gillings et al., 2008*; *González-Zorn and Escudero, 2012*).

Recombination reactions leading to cassette acquisition and reshuffling are governed by integron integrases (hereafter, *integrases*). Phylogenetic studies show that integrases belong to the family of tyrosine (Y)-recombinases, but form a discrete cluster within it, that evolved recently (*Mazel, 2006*; *Nunes-Düby et al., 1998*). Y-recombinases are a large family of DNA-recombinases. They typically

recognize specific sequences on double stranded (ds-) DNA substrates and recombine them following an archetypical pathway of two sequential strand exchanges (*Grindley et al., 2006*). Integrases have become a new paradigm in the field of recombination (*Escudero et al., 2016*) for structural and functional peculiarities that differentiate them from the rest of Y-recombinases. The most relevant one is that integrases perform asymmetric recombination reactions between double and single stranded sites. Indeed, when cassettes are integrated, their sites (*attC*) are recombined as single stranded (ss) hairpins (*Bouvier et al., 2005*; *Francia et al., 1999*; *Johansson et al., 2004*) while the site in the integron platform (*attI*) is canonically double stranded (ds) (*Figure 1b*). Recombination of ss- and ds-DNA is unique to integrases and is crucial from an evolutionary standpoint because these reactions are semiconservative. This allows for a bet-hedging strategy that minimizes the risk of incorporating cassettes to the array. Yet, it also poses many structural constraints to the recombination process (*Nivina et al., 2016*) such as rendering the typical second strand exchange of Y-recombinases deleterious, because it would linearize the chromosome (*Bouvier et al., 2005*; *Loot et al., 2012*). These constraints are solved in integrons through the acquisition of an extra α-helix (I2) within the catalytic core of the integrase protein (IntI) (*Messier and Roy, 2001*; *Figure 1c*). Crystallographic data have shown that the I2 helix has a specific biological function: it recognizes and docks a set of nucleotides that protrude from *attC* sites (the extra-helical bases) (*MacDonald et al., 2006*) causing a conformational change that impedes the second strand exchange, so that the junction is solved through an alternative pathway involving a replication event (*Loot et al., 2012*; *Figure 1b*).

Compelling examples of evolutionary innovation include the emergence of enzymatic activity de novo (*Clifton et al., 2018*; *Kaltenbach et al., 2018*), the evolution of regulatory networks (*Baker et al., 2012*), or the subcellular relocalization of enzymes through gene fusions (*Farr et al., 2017*). Still, good experimental models are not abundant (*Farr et al., 2017*) and a large wealth of evidence in the field comes from a few highly informative ones, such as antibiotic resistance genes (*Novais et al., 2010*; *Salverda et al., 2010*; *Figliuzzi et al., 2016*; *Jacquier et al., 2013*). Here, functional innovation results from changes in substrate specificity toward a new antibiotic molecule – a trait that can be strongly selected for in the laboratory – but the reaction is, overall, conserved. These models show that innovation often results from single mutations located at flexible regions in the periphery of the catalytic core of proteins (*Aharoni et al., 2005*; *Tokuriki et al., 2008*; *San Millan et al., 2016*). Such flexibility enhances evolvability because mutations can give rise to novel activities while having little impact on the main function of proteins (*Aharoni et al., 2005*; *Petrie et al., 2018*). Examples of innovation through larger rearrangements, such as the insertion or deletion of entire polypeptide sequences (*Pries et al., 1994*; *Park et al., 2006*; *Boucher et al., 2014*), or in other proteins such as RAG DNA-recombinases (*Zhang et al., 2019*) exist, but are very limited. Whether structural plasticity allows for the coexistence of novel and ancestral functions in these cases, deserves – in our opinion – to be investigated in more depth, and is the aim of this work. Integrases have expanded the type of substrate recombined (ds- and ss-DNA), the recognition pattern (structure and sequence) and changed the recombination pathway (double and single strand exchange). They are therefore an appealing molecular model to study evolutionary innovation in DNA processing enzymes (*Escudero et al., 2016*).

The I2 domain of integrases is necessary for the cassette capture activity (*Messier and Roy, 2001*), confers ss- recombination capabilities (*MacDonald et al., 2006*) and has not been found in any other protein in the databases. Hence, the most parsimonious explanation on how integron integrases have evolved, is that the acquisition of I2 in a Y-recombinase-like ancestor paved the way to the innovation of IntIs. Grafting of a 20-residue-long domain could, presumably, have entailed a strong destabilizing effect. Nevertheless, we recently proved that *bona fide* ancestral activity of integrases -including the second strand exchange resolution- is conserved on ds-DNA at low rates. This bi-functional state, and the lack of known function for the ancestral activity in contemporary integrons, strongly suggests that innovation of integrases was a smooth evolutionary process in which modern and ancestral activities were initially compatible to some extent (*Escudero et al., 2016*). Subsequent optimization events would have then turned the novel activity into the main one, in detriment of the ancestral activity.

Integrases are therefore a good experimental model to investigate if novel and ancestral functions can coexist when novelty emerges from large rearrangements (*Tawfik, 2006*). To explore this, in this work we have evolved the modern integrase IntI1 toward its ancestral function, the

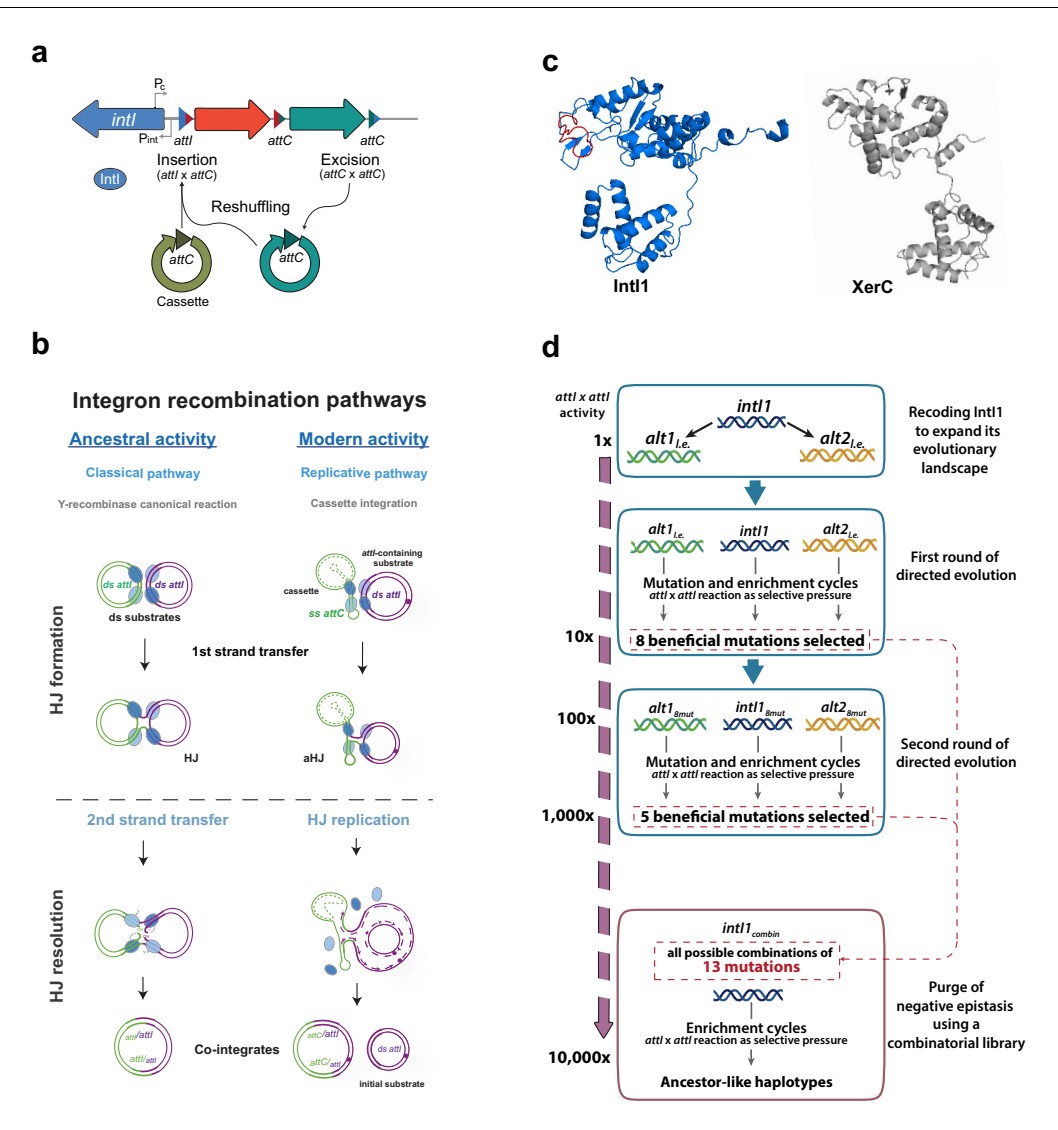

**Figure 1.** Introduction and outline of this work. (**a**) Diagram of an integron. The stable platform of integrons contains the integrase-coding gene (*intI*), the promoters dedicated to the expression of the integrase and the cassettes (P$_{int}$ and P$_c$, respectively), and the *attI* site (blue triangle). The variable part is composed of cassettes, encoding genes of different functions (arrows facing right) and their cognate *attC* sites (red and turquoise triangles). Recombining different sites, integrases can incorporate (*attI* x *attC*), excise (*attC* x *attC*), and reshuffle cassettes (excision followed by integration reactions) within the platform. (**b**) Recombination reactions in integrons can follow two distinct pathways: a replicative one when at least one substrate is an *attC* site -this is the *modern* activity of integrases-; or the classical pathway with a second strand exchange when recombining exclusively *attI* sites. This classical pathway is considered the *ancestral* activity of integrases for its resemblance with that of other Y-recombinases. (**c**) Structural models of *E. coli* XerC and Class one integron integrase IntI1, in which the I2 domain is marked in red. Models were obtained using Phyre2. d: Schematic representation of the setup followed in this work and the results (in form of mutations and gain in *attI* x *attI* activity) obtained at each step.

recombination of ds-substrates. With a succession of in vitro evolution experiments, using synonymous genes to extend the range of explored mutations (*Cambray and Mazel, 2008*; *Escudero et al., 2018*), we have identified 13 mutations increasing the ancestral function (*Figure 1d*). A combinatorial library allowed to filter out negative interactions and to obtain a combination of mutations with a 10,000-fold increase in recombination activity on ds-DNA. This increase came with an asymmetric trade-off (100-fold decrease) on ss-DNA recombination. Deep sequencing of this library revealed the epistatic and mechanistic making up of innovation in integrases. We show

that modern integrases encoding the I2 domain can reach high levels of ancestral activity, through only eight mutations, while retaining moderate levels of the modern one, strongly supporting that protein structural plasticity also fosters innovation through large rearrangements.

## Results

### Recoding the integrase gene into two alternative sequences

To explore a broader sequence space of IntI1 and obtain a larger set of possible beneficial mutations, we have used the Evolutionary Landscape Painter (ELP) (*Cambray and Mazel, 2008*). This algorithm enlarges the evolutionary landscape of a protein by recoding it using different codons. We recoded the IntI1 encoding gene (*intI1*) into two alternative genes that are synonymous to *intI1* but can evolve by exploring a different protein sequence space. After minor modifications aiming to decrease the stability of secondary structures in their transcripts, these low energy alternative genes, named *alt1*$_{l.e.}$ and *alt2*$_{l.e.}$, showed recombination activities comparable to those of *intI1* and were used as a starting point for directed evolution experiments (*Escudero et al., 2018*).

### Generating diversity and phenotype selection methods

In order to generate genetic diversity, upon which selection can operate, the three integrase-encoding alleles were further mutagenized using an error prone polymerase. For the *intI1* allele, we used a library of randomized *intI1* that had been previously constructed in our laboratory (*Demarre et al., 2007*). Both *alt*$_{l.e}$ alleles were randomized following a similar approach to produce libraries of approximately $7 \times 10^4$ clones for each allele (see materials and methods).

In order to enrich the population with beneficial mutations we need a selectable phenotype that results from recombination events. Initially, we adapted our previous method of generating plasmid cointegrates with antibiotic resistance markers (*Demarre et al., 2007*) to select for ancestral activity through the *attI* x *attI* reaction, and used it on *intI1* (*Appendix 1—figure 1*). We then developed a simpler and quicker method based on the *attI* x *attI*-mediated reconstitution of the *dapA* gene on the chromosome (*Escudero et al., 2016*) and used it with *alt1*$_{l.e}$, *alt2*$_{l.e}$ and *8xmut* alleles (see *Appendix 1—figure 1* and materials and methods).

### First round of directed evolution experiments

The library of *intI1* was selected for *attI* x *attI* efficiency using classical enrichment cycles for six cycles (*Appendix 1—figure 1a*), while alternative alleles were evolved using the new enrichment cycles for four cycles (*Appendix 1—figure 1b*). We then sequenced 18 to 25 clones of each library. Mutations found are listed in *Figure 2a*. Similarity between mutations found in *alt* and *intI1* libraries validated the new enrichment cycles (*Figure 2a*, substitutions marked in blue). Also, of the 15 different substitutions found in *alt* alleles, four were not reachable from the *intI1* gene through single nucleotide changes (*Figure 2a*, substitutions in red) confirming the exploration of a larger protein sequence space with alternative codes. Diversity in the *alt2*$_{l.e.}$ library was low, with all alleles containing D299E and A329T substitutions.

Recombination frequency for *attI* x *attI* was established for all relevant alleles (*Figure 2b*). Most alleles showed higher activities than their parental alleles, except for some haplotypes of the *alt1*$_{l.e}$ library, in which diversity was still high. Mutation T334S -found in *intI1*- did not confer a significant gain in recombination frequency, whether alone or combined with D161G, and hence represented a hitchhiking event (data not shown). To assess whether the increase in recombination rates was specific to the *attI* x *attI* reaction or a general gain in activity, we tested mutations found in *intI1* haplotypes for the *attC* x *attC* (cassette excision) reaction. All alleles showed lower recombination rates for this reaction than the WT (*Figure 3a*, 'one mut' strains), proving that the gain of function was specific for the ancestral reaction and showing a trade-off between activities.

### Construction of alleles pooling the eight mutations

From this first round of directed evolution experiments we selected eight different substitutions obtained in the three codes, conferring a gain in recombination activity (T118S, D161G, H162Q, K219R, D299E, G319E, G320D, and A329T) (*Figure 2a*, asterisks) and combined them together in the *intI1* allele to try to obtain a highly hyperactive mutant. Intermediate steps in the construction of

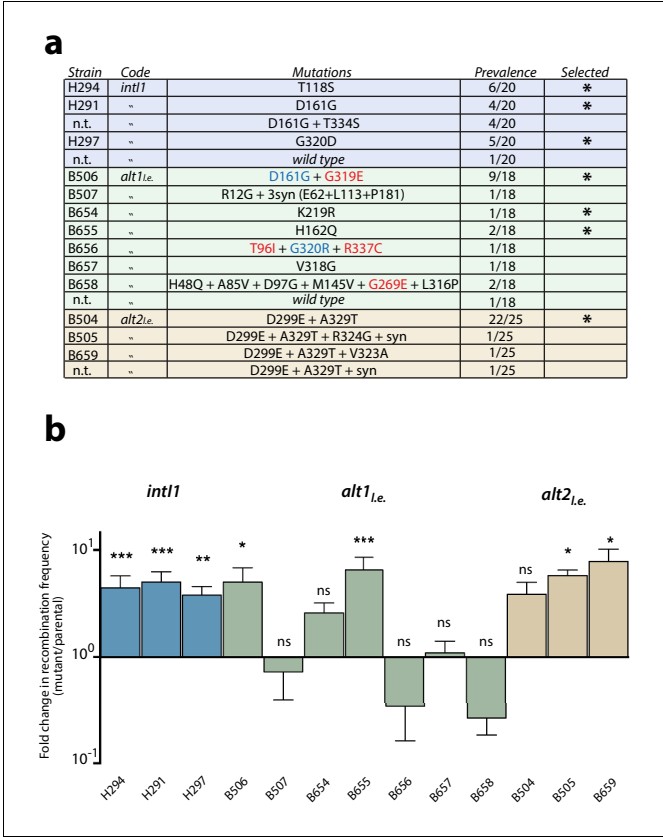

**Figure 2.** First round of enrichment cycles. Results are shown for *intI1* (blue) *alt1* (green) and *alt2* (brown). a: list of mutations found among the three alleles after enrichment cycles. Substitutions not reachable from the *intI1* code are highlighted in red. Substitutions marked in blue are similar to those obtained with *intI1*, validating the new selection cycles. Mutations selected for the next round are marked with an asterisk (n.t.: not tested). b: fold increase in *attI* x *attI* recombination of all haplotypes. Bars represent average values of at least three independent experiments. Error bars show standard error. Activity of each mutant was compared to that of the parental allele. Statistically significant differences are indicated by * (alpha = 0.05) and non-significance by 'ns'.

this mutant showed a clear trend toward *attI* x *attI* hyperactivity yet with some exceptions. For instance, the triple mutant (T118S, D161G, and G320D) is less active than any of the three possible double mutants. The final $intI1_{8mut}$ allele showed a 100-fold increase in ancestral (*attI* x *attI*) activity (from $1.82 \times 10^{-4}$ to $1.85 \times 10^{-2}$) compared to *intI1* (*Figure 3a*).

We sought to elucidate if a gain of recombination efficiency for the ancestral reaction came with a trade-off in the modern reactions, i.e.: *attI* x *attC* and *attC* x *attC* (*Figure 3a*). We observed a strong trade-off between *attI* x *attI* and *attC* x *attC* reactions (log-log regression $R^2$: 0.973; p<0.0001), and a weak 1 between *attI* x *attI* and *attI* x *attC* reactions (log-log regression $R^2$: 0.13; P: 0.034) (*Figure 3b*), suggesting that there is a conflict among protein conformations allowing integrases to specifically process *attI* or *attC* sites. Interestingly, trade-offs are not always homogeneous: for example, among $alt1_{I.e}$ and $alt2_{I.e.}$ alleles, some *attI* x *attI* hyperactive mutants show small gains for the *attI* x *attC* reaction too, while others show dramatic losses of activity (almost 6 orders of magnitude) (*Appendix 1—figure 2*). The variety of trade-offs suggest that there is a general -but not strict- balance between the structural conformations that allow for the efficient recognition or processing of *attI* and *attC* sites.

## Negative epistasis is prevalent among mutations

From the *attI* x *attI* data in *Figure 3a*, one can easily suspect that combination of beneficial mutations would perform less well than expected from the effect of individual mutations. This phenomenon, named negative epistasis, could hamper our efforts to obtain hyperactive integrases. To assess

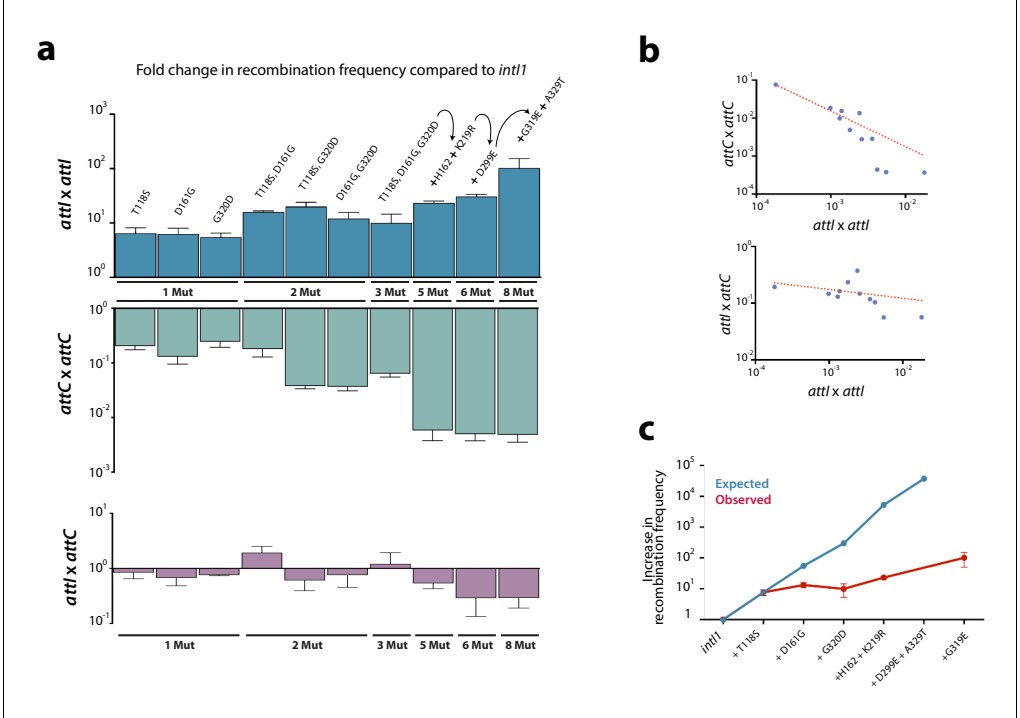

**Figure 3.** Construction of alleles with eight mutations. (a) fold change in recombination rates for *attI* x *attI*, *attC* x *attC*, and *attI* x *attC* of intermediate steps in the construction of *intI1_{8mut}* containing a variety of combinations of mutations. Bars represent average values of at least three independent experiments. Error bars show standard error. Mutations are indicated only on top of blue bars but mutants are depicted in the same order among the three graphs. Arrows and the '+' symbol in 5x, 6x, and 8x mutants mean that mutations shown are added to those in the previous mutant. Strain numbers are found in ***Supplementary file 2***. (b) negative correlation between the activity in *attI* x *attI* reactions and that in *attC* x *attC* (above) (Spearman test, r = −0.945, P (two-tailed) <0.0001; log-log regression $R^2$: 0.973); and *attI* x *attC* reactions (below) (Spearman test, r = −0.654, P (two-tailed) = 0.0336; log-log regression $R^2$: 0.13). (c) expected and observed increases in recombination rates among mutants containing subsets of the mutations in *intI_{8mut}*. Expected values are obtained based on a multiplicative model in the absence of epistasis. The lower recombination rates observed are indicative of negative epistatic interactions among mutations.

the presence of such epistatic interactions, we have plotted the *observed* and *expected* gains in recombination frequency for a subset of mutants for which we have data for isolated and combined mutations (***Figure 3c***). Observed recombination frequencies were clearly lower than expected (assuming a multiplicative model, typical of independent *loci*), suggesting pervasive negative epistasis in our experiments. As an example, the three mutations obtained with *intI1* in the first round of experiments (T118S, D161G, and G320D), conferring a 7.6-, 5.4-, and 7.2-fold increase in activity respectively, would lead in a non-epistatic scenario to a ≈300-fold activity increase. Instead, the gain in activity with the three mutations is approximately 10-fold and is actually slightly lower than the gain of double mutants (***Figure 3a***). The combination of these three mutations with all other mutations selected in the first round of experiments except G319E (for which we do not have individual data), would lead in a non-epistatic scenario to a 3.75·10^4-fold increase in activity in a theoretical *intI1_{7mut}*, again very far from the observed 10^2-fold gain obtained in the *intI1_{8mut}* bearing all mutations (***Figure 3c***). We obtained statistical support for negative epistasis by computing error in epistasis estimation ($\sigma_\varepsilon$) through error propagation (***Trindade et al., 2009***) (see Materials and methods). These negative epistatic interactions indicate that the benefits of these mutations are not independent. To go further we needed to explore the contribution of new mutations.

## Second round of directed evolution experiments

We sought to obtain the recombination rates on ds-DNA of integrase ancestors. Since we cannot determine this value, we take the rate of modern (ss-DNA) reactions in IntI1 as a hypothetical optimum. We hence aim to cover the four orders of magnitude gap between modern and ancestral activities in IntI1. We decided to re-evolve the *intI1₈mut* allele. To do so while preserving the exploration of a larger protein sequence space, we inserted the same mutations in both alternative alleles to obtain *alt1₈mut* and *alt2₈mut* and subjected them to the same re-evolution experiment in parallel. *intI1₈mut*, *alt1₈mut* and *alt2₈mut* alleles were subjected to randomization through error-prone PCR, established as libraries, and selected for *attI* x *attI* hyperactivity. After four enrichment cycles, we sequenced a subset of clones from each experiment and tested their recombination frequency through a modified chromosomal setup (as the one used in the new enrichment cycles) to increase the dynamic range of our experiment (see *Appendix 1—figure 3*, and Materials and methods). The level of variability within the population was still high at this point in all three populations, but in some cases focusing on the abundance of amino acid substitutions across haplotypes revealed a clearer result. The complete list of mutations found in this re-evolution experiment are listed in *Supplementary file 1*, but the main results are presented in *Figure 4a* and summarized below.

### intI1₈mut

Only two haplotypes, C326 and C325, were found more than once. Allele C326 contained, among others, a Y220N substitution within the I2 domain and showed recombination frequencies approximately seven times higher than the parental *intI1₈mut* haplotype and 400 times higher than WT *intI1* (*Figure 4b*). Haplotype C325 showed a toxicity phenotype, with cultures showing very poor growth

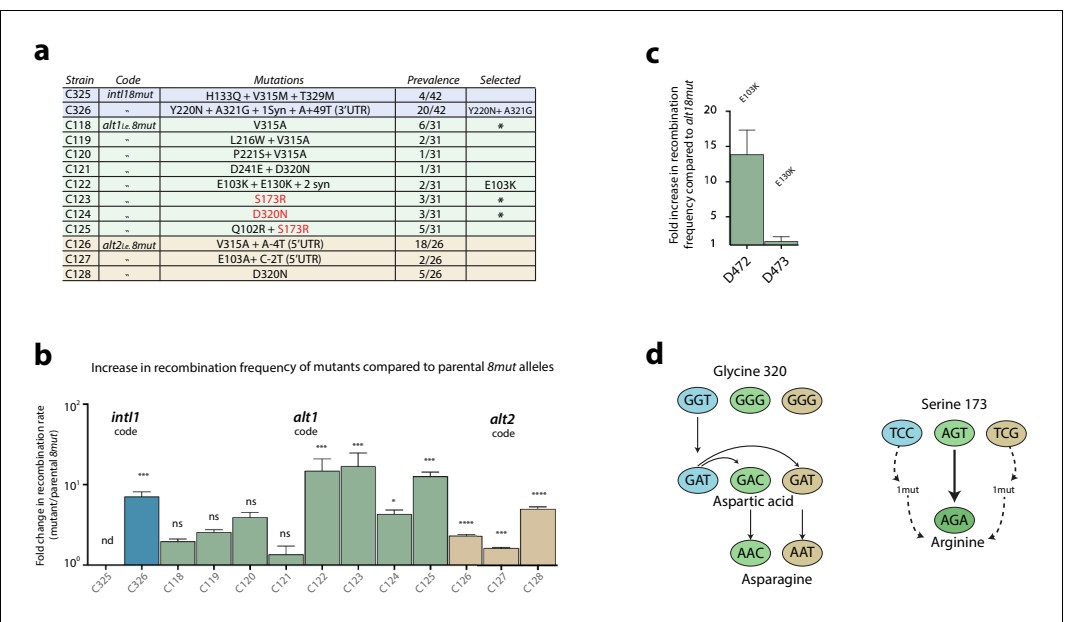

**Figure 4.** Second round of mutagenesis and enrichment cycles. a: list of mutations found among the three *8mut* alleles after four cycles. Substitutions that are not reachable from the *intI1₈mut* code are highlighted in red. b: fold increase in recombination rates for the *attI* x *attI* reaction of all mutants compared to their parental *8mut* allele. Bars represent average values of at least three independent experiments. Error bars show standard error. Statistically significant results are indicated by * (alpha = 0.05). c: Analysis of the mutations found in the *alt1₈mut* mutant C122. Of both non-synonymous mutations, only E103K produces a relevant increase in activity. d: Mutational pathways in our experiments highlight the importance of broad sequence space exploration. Mutation S173R, conferring a high activity gain, was only found in the *alt1* code because *intI1* and *alt2* codes needed an additional mutation to access that codon. Glycine to aspartic acid substitution (G320D) appeared in the *intI* allele in the first round of experiments and was included in both *alt₈mut* alleles using two different codons. It then re-evolved to asparagine (D320N), a residue that was not reachable from any of the starting codons with a single mutation.

in inducing conditions. Mutations were tested separately to decipher their contribution (*Appendix 1—figure 4*). How such an haplotype was present at this stage of purification of the library, remains cryptic.

## alt1$_{8mut}$

Diversity in this library was high (*Supplementary file 1*) but some amino acid substitutions were particularly prevalent among a variety of different haplotypes. V315A is, for instance, found in combination with other mutations in five other haplotypes (totaling 35% (11/31) of haplotypes). A similar phenomenon was observed for S173R and D320N substitutions. Prevalent mutations and alleles were tested, showing a variety of activity gains (*Figure 4b*). Mutations E103K and E130K were tested independently (*Figure 4c*), revealing that only E103K had a beneficial effect on activity. The re-evolution from G320D to D320N in this set of experiments underscores the intrinsic limitations of directed evolution experiments in sequence space exploration (*Figure 4d*). Interestingly, the S173R mutation -that is out of reach from the *intI1* code- exemplifies how the use of synonymous genes can partly alleviate such limitations.

## alt2$_{8mut}$

Sequencing of 26 clones revealed a more homogeneous population than *alt1$_{8mut}$* (*Figure 4a* and *Supplementary file 1*). The V315A substitution already observed in *alt1$_{8mut}$*, was the most prevalent one, followed by D320N and E103A/K. It is noteworthy that 20 haplotypes bore mutations in the 5'-UTR. These mutations do not change protein activity but may influence translation efficiency and hence bias our selection cycles. Here, the prevalence of 5' UTR mutations is high enough to suspect that this region is non-optimal. We assessed if the 5' region of the transcript could be interfering with transcription of the *alt2$_{8mut}$* gene (as we found in *Escudero et al., 2018*), but saw no differences in folding energies between WT and mutated 5'-UTRs ($-6.1$ kcal mol$^{-1}$ in all cases). This, combined with the linkage of UTR mutations to substitutions that appear alone in other codes, led us to consider UTR mutations in *alt2$_{8mut}$* as hitchhiking events.

Five mutations considered to be biologically relevant in this round of enrichment cycles, i.e.: E103K, S173R, Y220N, V315A, D320N, and A321G, were selected for further experiments.

## Purifying epistasis

Our results on the activity gain in mutants gathering several mutations suggested the presence of negative epistatic interactions between them (*Figure 3c*). In order to reach our goal of obtaining IntI1 mutants with a similar efficiency on *attI* x *attI* as that of WT IntI1 on *attI* x *attC* (modern integration reaction), we needed to obtain a four order of magnitude gain in activity. After two rounds of experiments some alleles showed increases of almost three orders of magnitude. Yet future experiments aiming to unravel structural aspects of integrase binding and processing of the *attI* site will need a highly active protein that is as close to the WT as possible. This, together with the negative epistasis observed, prompted us to avoid pooling together more mutations, but rather to try to obtain the minimal combination of mutations with the highest effect. To do so we established a library of mutants containing all possible combinations of 13 biologically relevant mutations. These are: E103K, T118S, D161G, H162Q, S173R, K219R, Y220N, D299E, V315A, G319E, G320D/N, A321G, and A329T. The rationale of this experiment is that by subjecting this library to our enrichment cycles we should be able to select the combination of mutations showing the highest levels of activity. Because of design constraints in regions with high variability (see Materials and methods) the theoretical number of different alleles in the library was 24,576. To maximize the chances of containing all alleles, we established a library of approximately five times this size. As a preliminary validation step, we sequenced 50 clones and observed a high level of heterogeneity of alleles, with an even distribution of WT and mutated sequences at each residue of interest. We also observed unintentional mutations in approximately 40% of the alleles. This was not completely unexpected, due to the number of rounds of PCR amplification performed to assemble all the fragments, and despite the use of low error rate polymerases (see Materials and methods). Once validated, we subjected this library to the new enrichment cycles to select for the best combination of mutations. An estimate of recombination rates was measured *en masse* at each step (*Appendix 1—figure 5*). Recombination activity increased 20-fold during the first four cycles and seemed to *plateau* from cycle five,

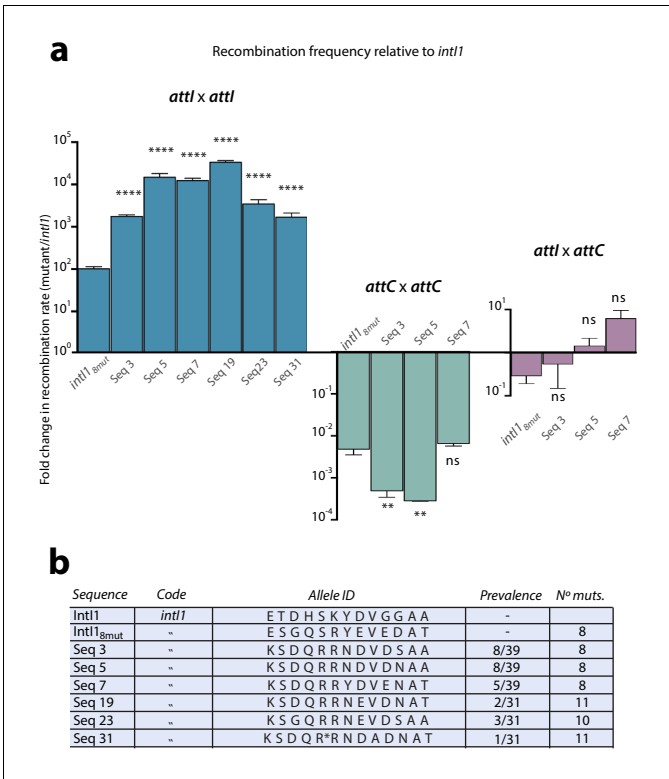

**Figure 5.** Epistasis purification. a: fold change in recombination rates -relative to *intI1*- for the *attI* x *attI*, *attC* x *attC,* and *attI* x *attC* reactions of mutants obtained after epistasis purification (*intI1*$_{8mut}$ is also shown for clarity). Gains in *attI* x *attI* activity entail asymmetrical losses for *attC* x *attC*. Bars represent average values of at least three independent experiments. Error bars show standard error. Statistically significant differences are indicated by * (alpha = 0.05) and non-significance by 'ns'. b: allele-ID (identity of the residues at the 13 positions of interest) of the most prevalent mutants after epistasis purification.

leading us to stop the selection process at cycle 6. We sequenced 39 clones of the library at this stage. Among them we obtained 18 different haplotypes of which 13 were unique and five were found eight, eight, five, three, and two times each. All sequences found more than once were tested for recombination frequency, together with one of the sequences with a single representative (*Figure 5a*). All clones showed significant increases in recombination rates compared to *intI1*$_{8mut}$, and three of them (Seq 3, Seq 5, and Seq 7) did so by combining a different subset of eight mutations (*Figure 5b*). Importantly, sequences Seq 5, Seq 7, and Seq 19 show a 100-fold increase compared to *intI1*$_{8mut}$, reaching our target of 10,000-fold increase compared to WT *intI1*. In these mutants, the activity levels for the *attI* x *attI* reaction are similar to those of the modern *attI* x *attC* activity in the wild type integrase. All alleles were specifically tested for second strand exchange activity (*Escudero et al., 2016*) and showed a variety of gains, with some reaching almost 2000 fold increase compared to WT *intI1,* confirming the increase in *bona fide* ancestral activity in some -but not all- cases (*Appendix 1—figure 6*) (see discussion). To assess the trade-offs entailed by these mutations for other reactions, sequences containing eight mutations were tested for the *attI* x *attC* and the *attC* x *attC* reactions in a similar chromosomal assay. Recombination activity was normalized to that of IntI1 and compared to the *intI1*$_{8mut}$ allele (*Figure 5a*). The asymmetry of trade-offs is in some cases remarkable. For instance, while the 10-fold increase obtained in Seq 3 for the *attI* x *attI* reaction comes with a symmetrical 10-fold loss for *attC* x *attC,* Seq 7 increases 100-fold its activity for the *attI* x *attI* reaction, while not decreasing at all its recombination rate on *attC* x *attC*. These alleles show a rising trend but no significant differences for the *attI* x *attC* reaction.

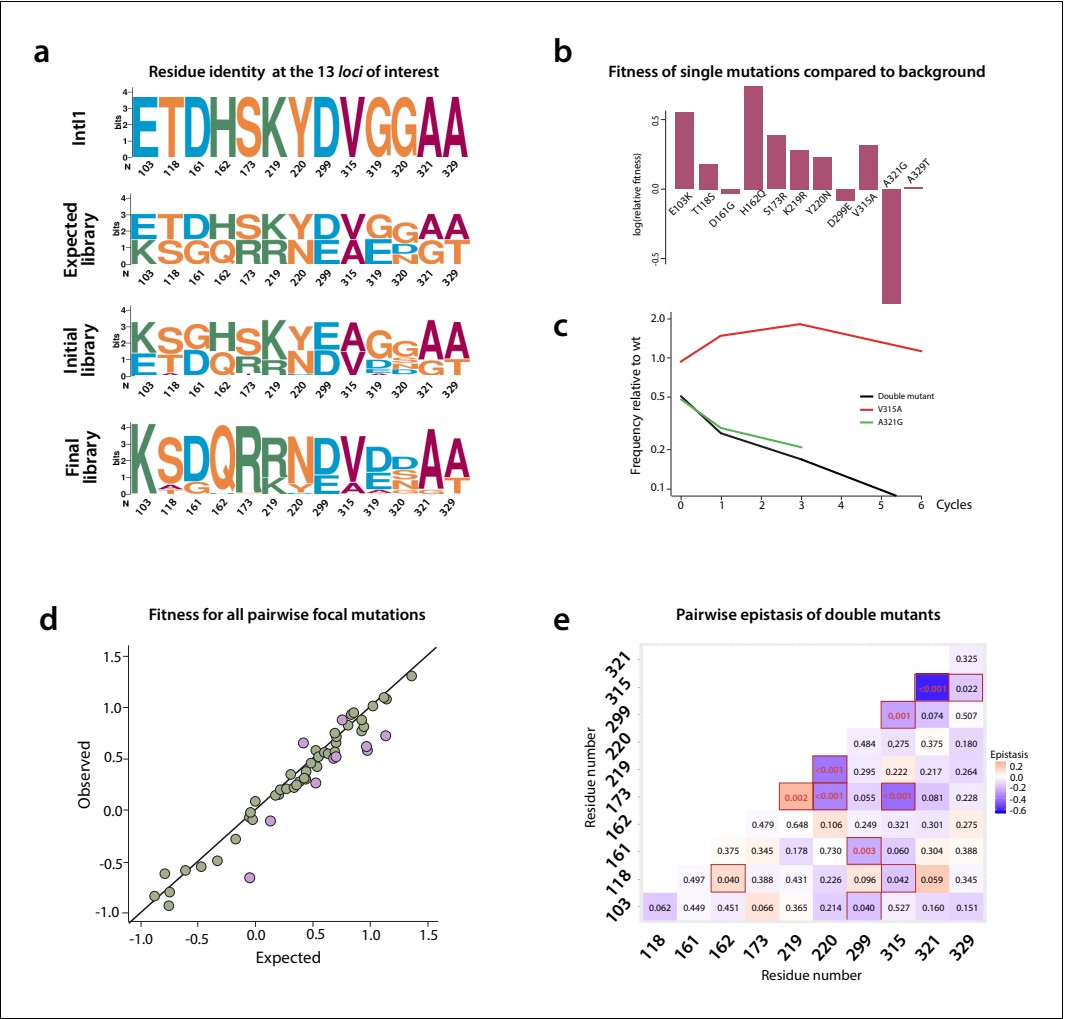

**Figure 6.** Mutation dynamics in the library. (**a**) logos showing allele-IDs reflecting the residue composition at each position of interest. Logos for the wild type protein (IntI1), the expected and observed composition of the library before the cycles and the final composition after six enrichment cycles. (**b**) fitness of single mutations compared to the background mutational composition of the library. (**c**) fitness effect of V315A and A321G mutations shows sign epistasis between both mutations. (**d**) fitness of double mutants. Plot representation of expected vs. observed fitness. The good linear correlation suggests a lack of diminishing returns among highly adaptive mutations. Purple dots represent double mutants in which epistasis values are statistically significant (red squares in panel e). (**e**) heat map representing epistasis between mutations in all positions of interest. P-values for each analysis are indicated within squares. Statistical significance is represented with red boxes (95% confidence interval) and red, bold numbers (99% confidence interval).

## Mutant dynamics during cycles

Understanding epistasis between the mutations obtained can reveal structural constraints influencing integrase evolution. To study epistasis interactions, we sequenced the initial library and three time-points of the epistasis purification assay (cycles one, three, and six) using PacBio SMRT bell sequencing. This technology provides long reads and allows phasing mutations into haplotypes.

Sequencing reads were aligned to *intI1* and the codons in the 13 loci of interest were extracted to obtain a string of amino acids that defines each allele. This allele-ID was used for further analysis for simplicity and because it circumvented the problems derived from the high error rate intrinsic to this technology (roughly $3 \cdot 10^{-3}$ in our experiments). The initial library was validated by the fact that the profile of the genetic background is very similar for all focal mutations (*Figure 6a* and *Supplementary file 4*). Unintended variability found in positions 319 and 320 at the initial library was due to the use of primers with two degenerate bases at those positions (see materials and

methods). It is therefore unlikely to be reflecting a biologically relevant phenomenon. We also observe a clear decrease in allele variability with time, confirming the selective pressure exerted during the cycles (*Appendix 1—figure 7*). Analysis of all mutations in the last time point showed a clear clustering in the 13 loci, justifying the use of allele-IDs (*Appendix 1—figure 7a*). The only exception was an unexpected accumulation of mutations leading to a A234V substitution that was initially included in our analysis, since we could not rule out its adaptive value.

We examined the effect of mutations by comparing the change over time in frequency of all haplotypes containing a particular *focal* mutation at a given locus, relative to all those containing the WT residue at that locus (*Figure 6b* and *Appendix 1—figure 7d*). This change in relative frequency is used as a proxy for fitness along our cycles (*Figure 6b*), and fitness is itself a proxy for recombination frequency. Mutations at three loci introduced a potential source of bias due to uneven linkage with mutations at the other *loci* of interest, and so had to be removed from the epistasis analysis: A234V (the mutation that appeared unintentionally) and all mutations at positions 319 and 320, which are especially complex (three and four possible residues respectively).

Despite these technical limitations (see materials and methods), our analysis reveals interesting epistatic interactions among mutations. All mutations but three (D161G, D299E, and A321G) are generally associated with a gain of fitness across all backgrounds (*Figure 6b*), and only A321G is not associated with a fitness gain in double mutants (*Appendix 1—figure 8*).

We can examine the pairwise epistatic tendencies of one particular mutation by computing the distribution of its fitness effects when paired with all other focal mutations (*Supplementary file 5*). Some mutations have narrow distributions, and therefore generally little pairwise epistasis with other focal mutations, while others show greater variance. An extreme case is V315A, which is beneficial when paired with all mutations except A321G, where it becomes deleterious. This is a case of sign epistasis, meaning that the V315A switches from beneficial to deleterious depending on the other mutations (*Figure 6b*). Both mutations are located in the C-terminal domain of the protein: V315A in the M helix and A321G between M and N helices. A321G that appeared in the 8mut background, is deleterious by itself and when paired with all other focal mutations but one (*Figure 6d and e*), although to highly varying extents. Interestingly, it shows positive epistasis with mutation T118S, located close to the F helix of the protein. Structural data from the *attC* x *attC* synaptic complex, showed that helix N buries one face in a hydrophobic pocket close to the F helix of the adjacent integrase monomer. This allows for the multimerization and stabilization of the synapse, providing a structural basis for this epistasis.

Mutations S173R and K219R also show strong positive epistasis. S173R is located in between the catalytic residue K174 and the R168 residue in charge of stabilizing *attC* EHBs, while K219R is located within the I2 domain that also stabilizes EHBs. Positive epistasis between these mutations likely reflects the important constraints that the need to recognize EHBs imposes on ds-DNA recognition and the trade-off between both activities.

## Discussion

Innovation is a key evolutionary phenomenon for which experimental data is scarce. In most experimental models, the structural basis for innovation are single mutations that increase protein promiscuity toward non-canonical substrates (*Aharoni et al., 2005*) while affecting only mildly the main protein function. These mutations are often located in the periphery of active sites, on surface loops that have great conformational flexibility (*Aharoni et al., 2005*; *Tokuriki et al., 2008*). Integron integrases (IntIs) belong to the Y-recombinase family of ds-DNA processing enzymes but have innovated toward a dual ss- ds-DNA recombination paradigm. This seems to have occurred through the acquisition of the 20 residue-long I2 domain within the core of the protein. A direct proof of causality between ss-recombination and the acquisition of I2 is not available, because its deletion leads to the complete loss of function of IntI1 (*Messier and Roy, 2001*). Yet, it has been clearly established that this domain has the distinct biological function of recognizing ss-DNA hairpins to avoid the deleterious second strand exchange on these substrates (*MacDonald et al., 2006*). This, together with the fact that both I2 and ss-DNA recombination are exclusively present in integron integrases, make the assumption that I2 allows for ss-DNA recombination the most parsimonious one. The fact that I2 has a distinct biological function contrasts with other examples of insertions within a protein catalytic core (*Boucher et al., 2014*), so it is non-trivial whether it was possible for its acquisition to provide

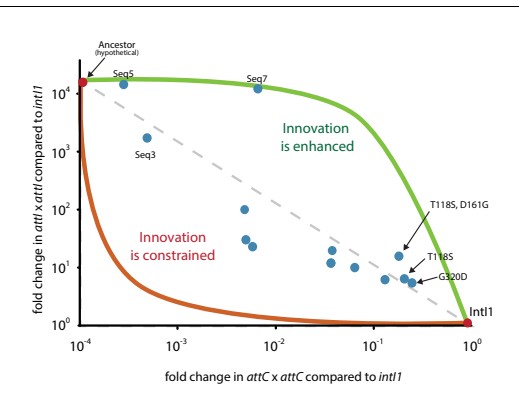

**Figure 7.** Mutant levels of ancestral and modern activities. Recombination frequency for *attI* x *attI* and *attC* x *attC* of mutants. Some haplotypes fall within the higher part of the graph, suggesting the possibility of assymetric trade-offs between functions fostering innovation.

substance ambiguity, without impacting negatively the main activity of the ancestor. Our mutants show that it is indeed possible, because the presence of I2 is compatible with high rates of ancestral and low rates of modern activity. Furthermore, a plot of modern and ancestral activity levels of our Seq5 and Seq7 mutants (*Figure 7*) shows that modern activity does not clearly drop despite the lack of selection for it during our cycles. Instead, some sequences (Seq5, Seq7, or the T118S+D161G double mutant) are good representatives of an evolutionary path fostering innovation, in which trade-offs between functions are asymmetric and proteins show substrate promiscuity. Hence, we show that innovation - whether resulting from single mutations or whole domain acquisitions- can follow the path of producing protein promiscuity without disrupting its main activity.

On top of this, our results have given us a deeper understanding on a variety of aspects of integrase activity.

## Structural insights

The crystal structure of *attC*-bound integrases has provided an unparalleled understanding on how I2 serves to recognize *attC* sites, with extrahelical bases acting as structural landmarks (*MacDonald et al., 2006*). The low affinity for the *attI* site shown by integrases in vitro (*Francia et al., 1999*; *Demarre et al., 2007*) has precluded obtaining crystals so far, limiting our understanding on the recognition and processing of *attI* sites. Mutations found here help understand the structural constraints of the ancestral reaction, and therefore the changes that integrases might have undergone during natural evolution. We find two significant regions where mutations accumulated: (i) the C-terminal domain where a high number of mutations mapped between the M and N helices. Helix N serves as an *anchor*, burying one face in a hydrophobic pocket of the adjacent integrase monomer in the synaptic complex. It is necessary for the multimerization and stabilization of the tetrameric synapse. Mutations between helices M and N probably entail changes in the flexibility of helix N producing a different conformation of the synapse that better suits the processing of the more rigid *attI* sites. (ii) The I2 domain that is the basis for innovation toward *attC* site- processing (*Nunes-Düby et al., 1998*; *Messier and Roy, 2001*; *MacDonald et al., 2006*) only accumulated two of the 13 mutations suggesting it is not the major hindrance for the ancestral activity in modern integrases.

The mutations found here, probably raise the ancestral activity through a variety of mechanisms, including an increase in the binding affinity to *attI,* and/or by stabilizing the *attI* x *attI* synapse. Both parameters are key for the formation of crystals, so we are currently using these mutants to obtain the long-awaited *attI*-IntI crystal structure. Interestingly, a general ability to deliver the second strand exchange seems to be independently selectable too. Indeed, we have obtained several alleles with approximately 10,000-fold increases in *attI* x *attI* activity (Seq 5, 7, and 19), but with large differences in double-strand exchange activity. No single mutation is clearly responsible for these differences, but the presence of Y220N (within the I2 domain) seems to correlate with higher activity levels.

It is also of note that we did not select any mutation in the first four α-helices in the N-terminal part of the protein, which form one side of the U-shaped clamp around the DNA target sites (*MacDonald et al., 2006*). This part of the recombinase is the most variable one among IntIs, and several amino acids from this segment of IntIA$_{Vch}$ make contact with the phosphate chain of structured *attC* sites (*MacDonald et al., 2006*). The lack of such mutations means that the way this segment accommodates the *attI* will remain unknown until the structure of the corresponding synaptic complex is obtained.

## Directed evolution experiments

The enrichment cycles used for directed evolution experiments have yielded examples of a variety of relevant evolutionary phenomena, such as the re-evolution of residue G320 to D and then to N (*Figure 4d*). This highlights the limits of directed evolution experiments, since the asparagine (N) was not reachable from any of the three initial codons and required a previous non-synonymous mutation (*Figure 4d*). It therefore vindicates our efforts to explore a larger sequence space while showing the limitations of doing so through synonymous mutations.

## Epistasis purification

Our combinatorial library of 13 mutations has proven to be an efficient method to avoid the constraints caused by negative epistatic interactions and improve protein activity, as demonstrated by the 100-fold increase in activity obtained while keeping the total number of mutations equal. The study of the behavior of mutations using deep sequencing has yielded interesting results despite strong technical limitations. For instance, the sign epistasis observed between V315A and A321G justifies by itself the need for a combinatorial approach to avoid those mutation combinations with detrimental epistatic effects. It is very likely that, had we not done this, the increase in recombination would have been mild due to sign epistasis between these mutations.

We studied epistasis systematically for pairs of mutations in our library, but could only detect it in a minority of cases (11/55, *Figure 6e*). When detected, epistasis was more often negative (nine cases with significant negative epistasis versus two with positive epistasis), suggesting that when there are interactions between pairs of mutations, the double mutant performs usually less well than expected. However, we did not clearly recover the classical global epistasis pattern of diminishing returns associated to fitness improvement (*Kryazhimskiy et al., 2014*). Usually, beneficial mutations have consistently smaller effects on fitter backgrounds. This absence could be the consequence of mutations acting on the different aspects of recombination activity (binding, multimerization, DNA-cleavage or rejoining...). Alternatively, it could also be due to our measure of fitness of single and double mutants, that is averaged across multiple genetic backgrounds. However, models predict that epistasis between pairs of mutations is limited when the adaptive process is initiated far from the optimum, as it is the case here (*Escudero et al., 2016*; *Blanquart et al., 2014*). In these conditions, epistatic interactions are barely detectable whether mutations are co-selected in an adaptive walk or selected independently and combined. However, when fitness gets closer to optimality, much more complex patterns of epistatic interactions are possible (*Blanquart et al., 2014*). Accordingly, the combination of more than two mutations suggests stronger epistatic interactions (*Figure 3c*) including some positive interactions (*Figure 6e*). Positive epistasis is the likely explanation to why S173R does not appear in the first round of experiments, despite the strong effect it has in the 8mut background; or why the generally deleterious A321G mutation appears in a background containing T118S.

## Limitations and other considerations

In this work, we have shown that we could shed some light on integrase innovation using a backward evolution approach, in which a modern protein is evolved toward its ancestral function. This has to be contrasted with the ancestral sequence resurrection strategy in which a putative ancestral sequence is inferred phylogenetically, synthesized and evolved toward a new function. Here, the process of ancestral sequence resurrection would be highly questionable because of long evolutionary time scales, the small number of IntI sequences available, and the lack of intermediates. Also, resurrection of an ancestral sequence and de novo insertion of residues is not always successful, because additional mutations elsewhere can be necessary (*Boucher et al., 2014*). In this situation, the use of experimental selection in the laboratory starting from a partially accurate resurrected ancestor (or from a close canonical tyrosine-recombinase relative) may be beyond reach. The strategy we used proved to be an efficient alternative that could be applied to other important switches in gene function that have occurred long ago. By strongly selecting for double-strand recombination, we have managed to show that I2 is compatible with high rates of ancestral activity and that a route through functional intermediates exists between integrases in an early innovative state and modern integrases. Still, lacking an ancestor sequence inferred phylogenetically, it is beyond our reach to determine whether this was the route naturally followed by integrases (in opposite direction), or the

impact that I2 had on the ancestor's activity. Also, we cannot rule out that the rise of new functions in IntIs actually occurred through gene duplications. Indeed, a plethora of accessory genetic elements in bacteria, such as plasmids, transposons or phage (integrated in tandem repeats), are naturally in a multicopy state that promotes the evolvability of their cargo genes (*San Millan et al., 2016*).

### Possible scenario for the emergence of integron integrases

Y-recombinases are abundant in phages, fomenting speculations about a viral (or partly viral) origin of integrons (*Escudero et al., 2016*). Interestingly, the CTX phage of *Vibrio cholerae* is integrated as a ss-DNA hairpin, but it differs from integron recombination in that it is mechanistically identical to ds-recombination. Indeed, the genome of CTXϕ mimics a ds-DNA molecule encoding a *dif* site to hijack the activity of the host's XerCD chromosome dimer resolvases -something done by many other elements dubbed IMEXs (integrative mobile elements exploiting Xer) (*Das et al., 2013*). Xer proteins are the closest relatives to IntIs (*Mazel, 2006*), but do not contain an I2 domain to recognize ss- substrates and avoid the second strand exchange that would result in a lethal linearization of the chromosome. To integrate safely, CTXϕ activates exclusively XerC, avoiding the deleterious cleavage of XerD. It is therefore possible that the acquisition of I2 occurred in an ancestor with Xer-like activity, and that it served initially to maintain the integrity of the chromosome when hijacked by ss-IMEXs that had not developed mechanisms to avoid the second strand exchange. In this scenario, a mildly negative impact of I2 on the main activity of the ancestor could have been compensated by the safer integration of such ss-IMEXs, that would otherwise linearize the chromosome and kill the host. Our results show that the ancestral integrase bearing I2 could have evolved without impairing the ancestral function up to a quite high activity on ss-DNA. From that detectable activity, natural selection could easily drive the specialization toward the new function, even at the cost of impairing the ancestral function as our data show (*Figure 7*).

### Concluding remark

The acquisition of an entire domain that is associated with a new function makes integrases an excellent experimental model for studying the evolution of novelty. Our results suggest that such a seemingly disruptive mutational event is compatible with a high ancestral activity, presumably thanks to the well-known robustness of the Y-recombinase fold (*Kwon et al., 1997*; *Tóth-Petróczy and Tawfik, 2014*). We further showed -see *Figure 7*- that an integrase with ancestral levels of ds-recombination activity and containing the I2 domain, could have detectable levels of activity on ss-DNA upon which selection can act. Using the *adaptive landscape* metaphor -in which genotype space is plotted on the x and y axis and fitness on the z vertical axis- our data suggest that the peaks representing the ancestral and modern functions are connected by some form of ridge that may facilitate the transition from one to the other, fostering innovation.

## Materials and methods

**Key resources table**

| Reagent type (species) or resource | Designation | Source or reference | Identifiers | Additional information |
|---|---|---|---|---|
| Strain, strain background (*Escherichia coli*) | DH5a | Lab strain | | supE44 DlacU169 (F80lacZ' DM15) DargF hsdR17 recA1 endA1 gyrA96 thi-1 relA1 |
| Strain, strain background (*Escherichia coli*) | UB5120 | PMID:2157593 | | F-pro met recA56 gyrA [NalR] |

*Continued on next page*

*Continued*

| Reagent type (species) or resource | Designation | Source or reference | Identifiers | Additional information |
|---|---|---|---|---|
| Strain, strain background (*Escherichia coli*) | β2163 | PMID:15748991 | | MG1655:: DdapA::(erm-pir) RP4-2-T c::Mu [KmR] |
| Strain, strain background (*Escherichia coli*) | B36 | PMID:26961432 | | MG1655 ΔdapA recA269::Tn10 attB::attl1WT-attl1STOP-dapA [SpR] |
| Strain, strain background (*Escherichia coli*) | 4137 | PMID:16341091 | | β2163/pSW23T:: attCaadA7 |
| Strain, strain background (*Escherichia coli*) | 2714 | PMID:15748991 | | β2163/pSW23T:: attl1 |
| Strain, strain background (*Escherichia coli*) | H203 | Strain from this work. Plasmid p6944 from PMID:19730680 | | β2163/p6944 (pSW23T:: attCaadA7-VCR2/1) |
| Recombinant DNA reagent | pBAD18 | PMID:7608087 | | pBAD18 |
| Recombinant DNA reagent | p6944 | PMID:19730680 | | pSW23T::[Ptac]-attCaadA7lacIq-VCR2-pir116* (BOT) |
| Recombinant DNA reagent | p2714 | PMID:15748991 | | pSW23T::attl1 |
| Recombinant DNA reagent | p929 | PMID:15716446 | | pSU38Δ::attl1 |

## Bacterial strains, plasmids, primers

The most important bacterial strains and plasmids in this work are listed above (key resources table) and were obtained elsewhere (*Escudero et al., 2016*; *Bouvier et al., 2005*; *Demarre et al., 2005*; *Martinez and de la Cruz, 1990*; *Guzman et al., 1995*; *Bouvier et al., 2009*; *Biskri et al., 2005*). A full list and more details on bacterial strains, plasmids and primers are described respectively in *Supplementary files 2* and *3*.

## General DNA procedures

Standard techniques were used for DNA manipulation and cloning (*Sambrook et al., 1989*). Restriction and DNA-modifying enzymes were purchased from New England Biolabs and Fermentas (Thermo-Scientific). PCR's (Polymerase Chain Reaction) were performed with Dreamtaq DNA polymerase, and Phusion polymerase (Thermo-Scientific) according to the manufacturer's instructions. 1% agarose electrophoresis gels were used to visualize DNA. DNA purification from PCR products and gels, as well as plasmid extractions, were performed using Qiagen kits. When necessary, DNA sequence was verified using an ABI BigDye Terminator v.3.1 sequencing kit and an ABI Prism 3100 Capillary Genetic Analyzer (Applied Biosystem). GATC and EUROFINS sequencing services were also used. Sequence analysis was performed using Geneious 7.1.

## Recoding the integrase into alternative genes

The Evolutionary Landscape Painter (ELP) (*Cambray and Mazel, 2008*), is an algorithm that recodes proteins into genes that are synonymous to the gene of origin, but have a different evolutionary landscape i.e.: they can reach parts of the protein sequence space that are not within the reach of the wt code. ELP maximizes the evolutionary potential of each amino acid while respecting codon

usage of *E. coli*. ELP has been recently updated to avoid the formation of secondary structures in the 5' region of the gene that interfere with transcription (*Escudero et al., 2018*).

## Sequence randomization

The *intI1* library of mutants used in this work had been established previously (*Demarre et al., 2007*). Using a similar approach, to produce diversity among $alt1_{l.e.}$, $alt2_{l.e.}$ and the three alleles containing eight mutations, we performed mutagenic PCR using the GeneMorph II kit from Stratagene. In the first set of enrichment cycles the amount of starting template (1 ng and 100 ng) and the number of PCR cycles (35) were adjusted to obtain, for each allele, two populations with approximate averages of two and six mutations per kb. We verified approximate mutation rates by sequencing a small subset of 24 clones from each population (1 ng $alt1_{l.e.}$=7.3 mut/gene; 100 ng $alt1_{l.e.}$=2.36 mut/gene; 1 ng $alt2_{l.e.}$=6.1 mut/gene; 100 ng $alt2_{l.e.}$=2.31 mut/gene). We mixed both populations to produce a single library of approximately $7 \times 10^4$ clones for each allele.

Given the small contribution to the experiment that highly mutated alleles had on the first round of experiments, we sought to decrease the mutation rate for the second set of cycles. We aimed to establish populations with averages of about two and four mutations per gene and obtained average mutation rates for $intI1_{8mut}$, $alt1_{8mut}$ and $alt2_{8mut}$ of 3.78, 1.5, 1.25 and 7.5, 4.18 and 4.72 mutations per gene. The size of the libraries was respectively of $8.7 \times 10^4$, $5.45 \times 10^4$, and $1.75 \times 10^6$.

## Enrichment cycles

In order to enrich our libraries with hyperactive integrases, selective pressure was applied upon all mutant libraries through a set of enrichment cycles. The cycles used for *intI1* (depicted in *Appendix 1—figure 1*) where based on those applied before (*Demarre et al., 2007*), with the only difference that the resident plasmid bore an *attI* recombination site instead of an *attC*. For *alt1*, *alt2*, all 8mut alleles and the epistasis purification experiment, we developed a new experimental setup that was streamlined to deliver rapid directed evolution cycles with lower background noise. To do so we used the DAP⁻ strain B36 that contains a chromosomal insertion of a *dapA* gene interrupted by two *attI1* sites in direct orientation (*Escudero et al., 2016*). Integrase-mediated recombination of these sites leads to the excision of the region between *attI1* sites and reconstitutes *dapA*, producing a selectable phenotype. Randomized alleles were cloned in a pBAD plasmid under the control of a $P_{BAD}$ promoter, electroporated in B36 and plated in media containing DAP and glucose to prevent the expression of the integrase. Colonies were then harvested from the plates into fresh media containing arabinose 0.2% and incubated at 37°C for one hour. After this induction time, the culture was plated on media without DAP and incubated at 37°C overnight. Colonies of recombinants were then pooled and used to extract plasmid content of this cycle that was electroporated into B36 again, starting the next cycle. This new setup was extremely rapid and efficient and hence, was used for the rest of the enrichment cycles.

## Recombination activity tests

In order to quantify the recombination activity of each allele we used two types of setups. First we used a suicide conjugation assay previously developed in our lab (*Biskri et al., 2005*; *Appendix 1—figure 3a*) and implemented in several occasions (*Nivina et al., 2016*; *Loot et al., 2012*; *Bouvier et al., 2009*). Briefly, an *attI* site is provided by conjugation carried on a suicide vector from the R6K- based pSW family that is known to use the Pir protein to initiate its own replication. This plasmid also contains an RP4 origin of transfer (*oriT*RP4). The donor strain β2163 carries the transfer functions in its chromosome, requires DAP to grow in rich medium and can sustain pSW replication through the expression of a chromosomally integrated *pir* gene. The DH5α recipient strains, which contain a pBAD plasmid [Ap^R] containing the allele to be tested and the resident plasmid pSU38Δ::*attI1* [Sp^R] carrying the *attI1* site, lacks the *pir* gene and therefore cannot sustain replication of the incoming *attI*-containing pSW vector. The only way for this vector to be maintained in the recipient cell is to form a cointegrate by *attI* x *attI* recombination (or *attI* x *attC* where applicable). The recombination frequency is calculated as the ratio of transconjugants expressing the pSW marker [Cm^R] to the total number of recipient clones [Ap^R, Sp^R]. Recombination frequencies correspond to the average of at least three independent trials. In the *attI* x *attC* tests performed to measure the trade-off between functions, plasmid pSU38Δ::*attI1* was substituted for plasmid pSU38Δ::$attC_{aadA7}$. For the

*attC* x *attC* reaction, a suicide pSW plasmid containing the *pir* gene interrupted by two *attC* sites is conjugated into a recipient strain containing the pBAD plasmid with the integrase allele to test. Recombination between both sites leads to the reconstitution of *pir* so that the plasmid can sustain its own replication (*Bouvier et al., 2009*). Recombination frequency is calculated as above.

After the first round of enrichment cycles that yielded the eight mutations that combined provided a 100-fold gain in recombination activity, we needed to enlarge the dynamic range of this test to be capable of measuring recombination frequencies of mutants to come. To do so, we used the same recombination system used for the new enrichment cycles in which integrase-mediated recombination of two chromosomal *attI1* sites reconstitutes *dapA* and produces a selectable phenotype (*Escudero et al., 2016*; *Appendix 1—figure 3a*). Integrase-coding alleles were expressed from a pBAD promoter for three hours using arabinose 0.2% before plating in media with and without DAP. The ratio between the number of colonies growing without DAP (recombinants) and those growing in DAP-containing media (total) gives the recombination frequency of that allele. In this new setup, recombination rates are 100-fold lower than in conjugation experiments, and we also have one more logarithm of dynamic range since conjugation is no longer limiting (*Appendix 1—figure 3b*).

## Mutant selection

We selected mutants following an evolutionary rationale to the interpretation of our results, and considered to be relevant those mutations that appear many times in the same or in different codes, alone or in combination with others, despite producing increases in recombination activity that were sometimes not statistically significant (for instance V315A).

Mutations in untranslated regions do not influence the intrinsic activity of the protein and are therefore not relevant for the evolution of the integrase. Yet, they can influence translation rates and hence play a role in the selection cycles. We therefore tested some to assess their influence in the recombination frequency of the allele obtained (as we have done for *intI1_{8mut}* mutants (*Appendix 1—figure 4*)). Notably, in the second round of enrichment cycles, the number of *alt2_{8mut}* alleles bearing mutations in the 5'UTR of the gene is high enough to be potentially interpreted as if this region was non-optimal. We hence explored the possibility that the 5' folding of the transcript was interfering with transcription of the *alt2_{8mut}* gene (as we saw in *Escudero et al., 2018*), but folding energies were identical between WT and mutated 5' UTRs ($-6.1$ kcal mol$^{-1}$ in all cases). Here, using the evolutionary rationale, and on the basis of frequency of appearance, UTR mutations can most often be considered hitchhike events (and neglected) i.e.: UTR mutations are not present in other mutants from *intI1_{8mut}* or *alt1_{8mut}*, and when they appear in *alt2_{8mut}* they are combined with mutations that are prevalent in the other codes. Hence, such parallel evolution events are considered as biologically relevant and help guide our choice of relevant mutations.

## Calculation of epistasis error

Error in epistasis ($\sigma_\varepsilon$) was calculated through the error propagation method as described in *Trindade et al., 2009*. The expected minus the observed fold increases in recombination frequencies (the epistasis) were 41.9; $2.9 \cdot 10^2$ and $5.24 \cdot 10^3$ for the double, triple and quintuple mutants respectively. Error values were 18.4, 41.1, and 108.1, respectively, proving that epistasis was significative.

## Epistasis purification library

To construct a library with all possible combinations of our 13 mutations we designed oligonucleotides containing a mixture of bases at each position of interest. Several fragments bearing the mutations were amplified and assembled in an approach that combined SOE-PCR splicing through overlap extension PCR (*Heckman and Pease, 2007*) and Gibson assembly (*Gibson et al., 2009*). Design constraints were found in more variable regions that introduced more variants. For instance, in position 319, in order to obtain WT (glycine) and mutated (glutamate) alleles we used primers with the codon GRM, that can indeed lead to the residues glycine (GGA, and GGC) and glutamate (GAA), but also to the unintended aspartate (GAC). Also, in the case of the re-evolved G320(D,N) residue, we used primers with an RRT codon that leads to all three residues (glycine: GGT; aspartate:

GAT; asparagine: AAT) but also to a serine (AGT). These constraints rendered the library more complex and raised the theoretical number of different alleles in it to 24.576.

Despite the use of high fidelity polymerases at all stages, the several rounds of PCR amplification needed along the process meant that we could not rule out a certain degree of randomization in the final library.

## Sequencing

Deep sequencing was performed using PacBio SMRTbell technology (The Genome Analysis Centre, Norwich, UK). PacBio technology produces long sequencing reads with albeit with high error rates (10%). The inclusion of single stranded loops at the edges of PCR-amplified alleles, yields long reads with several copies of a unique allele. This technology strongly decreases the error rate per allele. In our sequencing runs error rates were roughly estimated to be around 0.3%. Although the fidelity was significantly higher, this rate still implies an average of three mutations per allele (1014pb), precluding the use of complete allele sequences for the study. In an effort to capture the diversity of the initial library we dedicated eight SMRT Cells (theoretical number of reads: $4 \times 10^5$) for time 0. To observe the evolution of the alleles along the different cycles we dedicated two SMRT cells for each time point (cycles one, three, and six. Theoretical number of reads per time point: $1 \times 10^5$). Sequencing results underwent a curation process. First, reads that were too short to contain all mutations were discarded. Then reads were aligned to *intl1* revealing a high abundance of indels in our reads. This was also the case for reads from cycle six, where inactive integrases are extremely unlikely to be found, suggesting that indels were rather a technical artifact than a biologically relevant phenomenon. Hence, indels were artificially corrected in our reads. The combination of residues encoded in positions of interest (those in which we had inserted a mutation) was extracted for each allele, conforming a 13-letter string defining that allele: the alleleID. The total amount of usable alleleIDs after curation of our sequencing data was of 67179 reads for the initial library, and 21622, 31694, and 18907 for cycles one, three, and six. The risk of using allele-IDs, is underestimating the effect of mutations in positions other than the 13 *loci* of interest, that could have appeared during the process of construction of the library, and that could have been non-neutral. To address this, we analyzed the number of mutations found at each position in cycle six (*Appendix 1—figure 7A*). Mutations clearly clustered in the 13 *loci* of interest and were distributed homogeneously along the rest of the gene at a lower frequency, except for an unexpected accumulation of mutations leading to a A234V substitution, that was initially included in our analysis.

The current trade-off in sequencing technologies between long reads and coverage has meant that despite our effort to establish fully-defined alleles by phasing all mutations, coverage is too low to perform statistically robust analysis on the level of these complete alleles. We hence examined the effect of mutations by comparing the change over time in frequency of all alleles containing a particular *focal* mutation at a given *locus* of interest, relative to all those containing the WT residue at that *locus* (*Figure 6a* and *Appendix 1—figure 7D*). This change in relative frequency is used as a proxy for fitness along our enrichment cycles, and fitness is itself a proxy for recombination frequency. This analysis, in which the effect of a mutation is effectively measured as its average effect over all genetic backgrounds (albeit weighted according to the precise composition of the initial library), provides an interesting measure of mutation effects from both an evolutionary and engineering perspective. It is justified by the fact that the profile of the genetic background in the initial library is very similar for all focal mutations (*Supplementary file 4*).

To find epistatic interactions between mutations, we focused on the first two time-points because the profile of the genetic background for each mutation changes with time, influencing the apparent fitness of mutations and precluding proper analysis. To assess the significance of our epistasis estimates in the face of the counting noise inherent in deep-sequencing experiments, we simulated our experiment by starting with the observed genotype counts from the initial library and using the fitness estimates obtained for single mutations to compute genotype counts after cycle one, assuming independent mutation effects (no epistasis). The new counts were computed by Poisson sampling, using the observed total counts for cycle one, and apparent epistasis was calculated from this simulated data as for the real data. The results from multiple no-epistasis simulations could then be confronted with our real epistasis estimates (*Figure 6c,d and e*).

## Data processing

In order to analyze epistatic interactions between mutations we removed all sequences that were not WT at positions 319 and 320, and all sequences that were not WT or the expected mutant at all other positions.

## Fitness estimation

The limited read coverage of mutant sequences constrained us to measuring the fitness effects of focal mutations averaged over all backgrounds present in the library, rather than specifically in the WT background. Thus, the relative fitness effect of a particular mutation (or mutation pair) is defined as the collective fitness of all sequences carrying the mutation (or mutation pair) relative to the fitness of all sequences who are WT at the relevant position(s). This in fact provides a more general measure of fitness which takes into account a mutation's average epistasis with the genetic background of our library, rather than just focusing on its effect in a single background (*ie*. the WT background). This approach is permitted by the common profile of the genetic background for all single mutations and mutation pairs at $t_0$, as well as for their respective WT-containing reference sequences (*Supplementary file 4*), and by the sufficient read coverage at this level of diversity (for single mutations: 910x at $t_0$ and 127x at $t_1$; for double mutations: 182x at $t_0$ and 25x at $t_1$). However, only $t_0$ and $t_1$ could be used for the approach to be valid, as the profile of the genetic background of each focal mutation can change differently over time, due indeed principally to epistasis between focal mutations and the genetic background.

As only a single selection cycle is taken into account, the relative fitness effect, *w*, of a focal mutation, *i*, is computed as:

$$W_i = \text{In}(mut_i^{t1}/mut_i^{t0},)$$

here $mut_i^{t0}$ and $mut_i^{t1}$ are the ratio of mutant *i* counts to the relevant WT reference counts (those sequences with a WT residue at position *i*) at $t_0$ and $t_1$, respectively. The same principle is applied for calculating the relative fitness effect of mutation pairs. Pairwise epistasis, $\varepsilon$, between focal mutations, *i* and *j*, is computed as:

$$\varepsilon_{ij} = W_{ij} - (W_i + W_j),$$

where $w_{ij}$ is the relative fitness of the double mutant.

## Simulation of selection-sequencing experiment as a null model of pairwise epistasis

To assess the significance of our pairwise epistasis measures, we simulated the selection-sequencing experiment under the assumption of no epistasis to see how much of the measured epistasis might be due not to true fitness interactions but to errors in fitness measurement. The main source of noise was assumed to be sampling noise arising from the limited sequencing read coverage (any sampling noise arising from the selection-enrichment protocol itself was neglected, due to the maintenance of a large population size relative to mutant diversity). Other sources of noise, such as off-target mutations, are outside of our reach, and so the epistasis confidence estimates derived here serve as an upper bound.

Simulations were initiated with the real count data for $t_0$, and every genotype was assigned a relative fitness, *w*, based on the real fitness estimates for single focal mutations and the assumption of no epistasis between mutations (unlike above, *w* is defined here on a linear, rather than ln, scale). Poisson sampling, based on the real total sequencing reads for $t_0$, was performed to simulate true genotype frequencies at $t_0$. $t_1$ genotype frequencies were then generated according to: $f_a^{t1} = f_{at}^{t0}$ x $w_a/\bar{w}^{t0}$, where $f_a^{t0}$ and $f_a^{t1}$ denote the frequency of genotype *a* at $t_0$ and $t_1$, respectively, $w_a$ is the relative fitness of genotype *a*, computed as described above, and $\bar{w}^{t0}$ is the mean fitness of all genotypes at $t_0$. Finally, $t_1$ counts were generated from the simulated $t_1$ frequencies by an additional Poisson sampling, based on the real total sequencing reads for $t_1$. The simulated $t_1$ counts could then be used along with the real $t_0$ counts to derive fitness estimates for all single focal mutations and mutation pairs, and thus pairwise epistasis estimates for all mutation pairs, exactly as for the real data. 1000 simulations were performed, and the resulting distributions of pairwise epistasis estimates were compared directly with the real pairwise epistasis estimates to assign them confidence

values (simply the frequency at which the simulated estimates are less than, or greater than, the real estimate, for positive and negative epistasis, respectively) (*Supplementary files 5* and *6*).

## Acknowledgements

This work was supported by the Institut Pasteur, the Centre National de la Recherche Scientifique (CNRS-UMR3525), the European Union Seventh Framework Program (FP7-HEALTH- 2011-single-stage), the 'Evolution and Transfer of Antibiotic Resistance' (EvoTAR) project, the FP7-FET Proactive 'Plaswires' project; the Fondation pour la Recherche Medicale project DBF20160635736, the French Government's Investissement d'Avenir program Laboratoire d'Excellence 'Integrative Biology of Emerging Infectious Diseases' (ANR-10-LABX-62-IBEID) and the French National Research Agency (ANR-12-BLAN-DynamINT). JAE is supported by the European Research Council (ERC) through a Starting Grant (grant agreement n° 803375), the *Atracción de Talento* Program of the *Comunidad de Madrid* (2016-T1/BIO-1105), the *Ministerio de Ciencia, Innovación y Universidades* (BIO2017-85056-P) and the Marie Curie Intra-European Fellowship for Career Development (FP-7-PEOPLE-2011-IEF, ICADIGE). AN is supported by Paris Descartes University – Sorbonne Paris Cité, Fondation pour la Recherche Medicale (FDT20150532465) and École Doctorale Frontières du Vivant (FdV). OT and HK were supported by the European Research Council under the European Union's Seventh Framework Programme (ERC grant 310944) and OT received funding from Grant Equipe Fondation pour la Recherche Médicale EQU201903007848. We are very grateful to Álvaro San Millán and Craig MacLean for critical reading of the manuscript.

## Additional information

### Funding

| Funder | Grant reference number | Author |
|---|---|---|
| Centre National de la Recherche Scientifique | CNRS-UMR3525 | Didier Mazel |
| H2020 Marie Skłodowska-Curie Actions | PIEF-GA-2011-303022 | José Antonio Escudero |
| EU FP7 HEALTH | 282004 | Didier Mazel |
| EU-FP7 FET | 612146 | Didier Mazel |
| Fondation pour la Recherche Médicale | DBF20160635736 | Didier Mazel |
| Agence Nationale de la Recherche | ANR-10-LABX-62-IBEID | Didier Mazel |
| Agence Nationale de la Recherche | ANR-12- 897 BLAN-DynamINT | Céline Loot |
| European Research Council | StG-803375 | José Antonio Escudero |
| Comunidad de Madrid | 2016-T1/BIO-1105 | José Antonio Escudero |
| Ministerio de Ciencia e Innovación | BIO2017-85056-P | José Antonio Escudero |
| Fondation pour la Recherche Médicale | FDT20150532465 | Aleksandra Nivina |
| Fondation pour la Recherche Médicale | EQU201903007848 | Olivier Tenaillon |
| European Union 7th Framework Programme | 310944 | Harry E Kemble Olivier Tenaillon |
| Pierre and Marie Curie University | FP-7-PEOPLE2011-IEF | José Antonio Escudero |
| Pierre and Marie Curie University | ICADIGE | José Antonio Escudero |

The funders had no role in study design, data collection and interpretation, or the decision to submit the work for publication.

### Author contributions

José Antonio Escudero, Conceptualization, Data curation, Formal analysis, Validation, Investigation, Visualization, Methodology, Writing - original draft, Project administration; Aleksandra Nivina, Harry E Kemble, Data curation, Formal analysis, Investigation, Methodology, Writing - original draft; Céline Loot, Conceptualization, Data curation, Formal analysis, Supervision, Validation, Investigation, Methodology, Writing - original draft; Olivier Tenaillon, Data curation, Formal analysis, Supervision, Writing - original draft; Didier Mazel, Conceptualization, Resources, Supervision, Funding acquisition, Validation, Investigation, Methodology, Writing - original draft, Project administration

### Author ORCIDs

José Antonio Escudero (iD) https://orcid.org/0000-0001-8552-2956
Aleksandra Nivina (iD) http://orcid.org/0000-0002-1802-3724
Harry E Kemble (iD) https://orcid.org/0000-0003-1480-5873
Céline Loot (iD) https://orcid.org/0000-0003-3774-249X
Olivier Tenaillon (iD) http://orcid.org/0000-0002-3796-1601
Didier Mazel (iD) https://orcid.org/0000-0001-6482-6002

### Decision letter and Author response

Decision letter https://doi.org/10.7554/eLife.58061.sa1
Author response https://doi.org/10.7554/eLife.58061.sa2

# Additional files

### Supplementary files

• Supplementary file 1. mutations found in enrichment cycles.

• Supplementary file 2. bacterial strains used in this work.

• Supplementary file 3. plasmids and oligonucleotides used in this work.

• Supplementary file 4. Distribution -in single and double mutants- of mutations in the rest of loci of interest.

• Supplementary file 5. Observed and simulated values of fitness distribution for single mutants.

• Supplementary file 6. Histograms representing, for every mutation pair, the apparent epistasis measured for simulated data (for 1000 simulations in the absence of epistasis). Red lines show epistasis measured from the real data.

• Transparent reporting form

### Data availability

Sequencing data has been deposited in Dryad under accession code https://doi.org/10.5061/dryad.zcrjdfn7x.

The following dataset was generated:

| Author(s) | Year | Dataset title | Dataset URL | Database and Identifier |
|---|---|---|---|---|
| Escudero JA, Mazel D | 2015 | Raw sequencing data | https://doi.org/10.5061/dryad.zcrjdfn7x | Dryad Digital Repository, 10.5061/dryad.zcrjdfn7x |

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

## Appendix 1

**Appendix 1—figure 1.** Diagram of the enrichment cycles used to select for integrase variants hyperactive for *attI x attI*. (**a**) Cycles adapted from *Demarre et al., 2007* used for the evolution experiments with *intI1*. Briefly, a randomized library of integrase encoding genes cloned in a pBAD plasmid (**A**) is established in a toxin-resistant *E. coli* strain containing a pSU plasmid that encodes an *attI* site and a toxin (1). This strain acts as receptor in the conjugation of an *attI*-bearing suicide plasmid (2). Mutagenized integrases deliver the *attI x attI* recombination reaction allowing the formation of a selectable cointegrate (3). pBAD plasmids are then purified in two steps: the plasmid preparation from recombinants (4) is digested to specifically degrade cointegrates (5) and then transformed in a toxin sensitive strain (6). Plasmid extraction from this strain yields very pure pBAD plasmids containing a variety of integrase-coding genes of high activity (**B**). Higher selective pressure is applied by subjecting these plasmids to further cycles. b: Novel enrichment cycles used for the evolution of the rest of alleles. Briefly: the library of integrase mutants cloned in a pBAD plasmid (**A**) is established in a *dap- E. coli* that contains a *dapA* gene in the chromosome interrupted by two *attI* sites (1). Expression of the integrase leads to the recombination of both sites and the reconstitution of *dapA*, allowing recombinants to grow in media not supplemented with DAP (2). Plasmid preparations from recombinants (3) yields pure pBAD plasmids containing a mixture of hyperactive-integrase-coding genes (**B**) that can be further used in subsequent cycles.

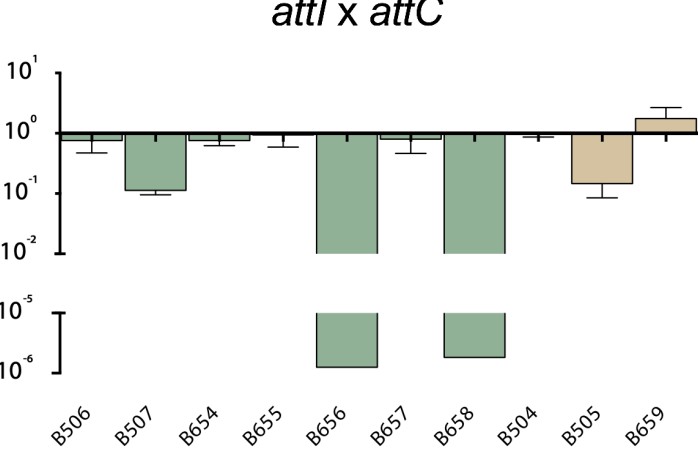

**Appendix 1—figure 2.** Recombination activity for the *attI* x *attC* reaction of evolved alleles from alternative codes, relative to their parental *alt* allele.

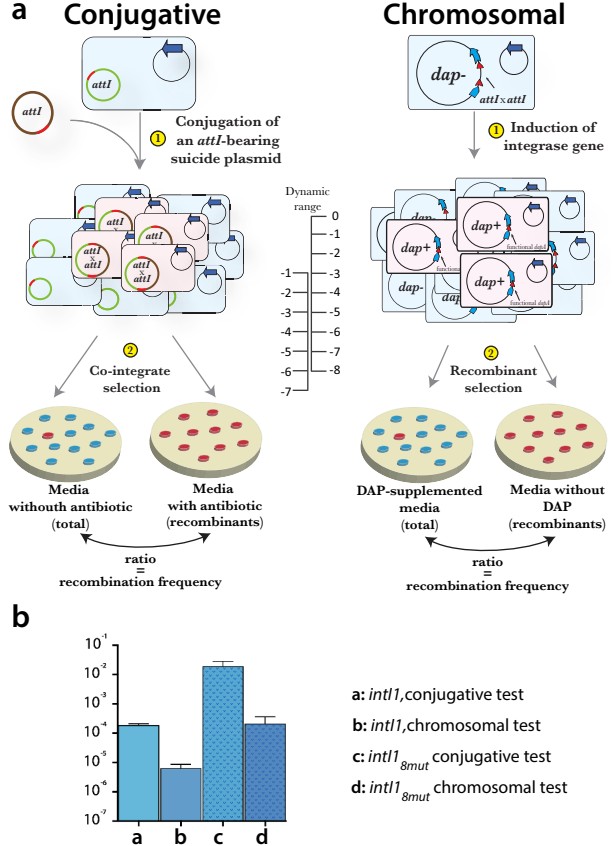

a: *intI1*, conjugative test

b: *intI1*, chromosomal test

c: *intI1*$_{8mut}$ conjugative test

d: *intI1*$_{8mut}$ chromosomal test

**Appendix 1—figure 3.** Methods used for evaluating the efficiency of integrase alleles. (**a**) scheme of conjugative and chromosomal tests. In both cases the recombination between *attI* sites produces a selectable phenotype. The dynamic range of each assay is different and so are the results for a given integrase allele. There is a drop of approximately two orders of magnitude between the conjugative and the chromosomal test. The chromosomal assay allowed us to determine the efficiency of highly active mutants that were close to the upper limit ($10^{-1}$) of the conjugative test. (**b**) example of recombination rates for *intI1* and *intI1*$_{8mut}$ measured in both assays.

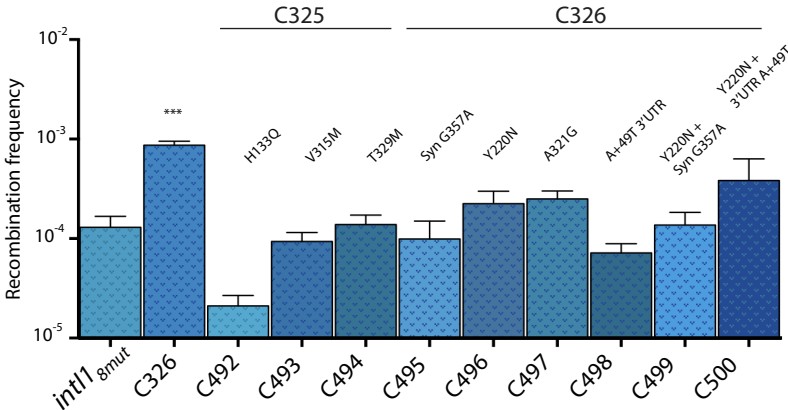

**Appendix 1—figure 4.** Recombination activity for the *attI* x *attI* reaction of subsets of mutations found in alleles C325 and C326.

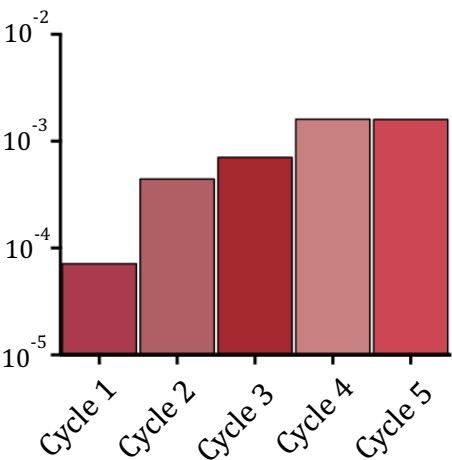

**Appendix 1—figure 5.** Increase in recombination rates for the epistasis purification library. Note that for technical reasons these data are not comparable to single mutant measurements.

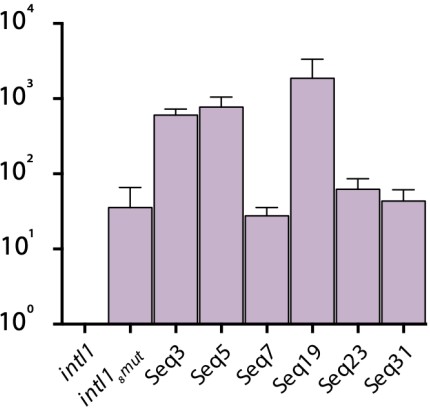

*Appendix 1—figure 6 continued*

**Appendix 1—figure 6.** Fold increase in double-strand exchange recombination of mutants from the epistasis purification experiment.

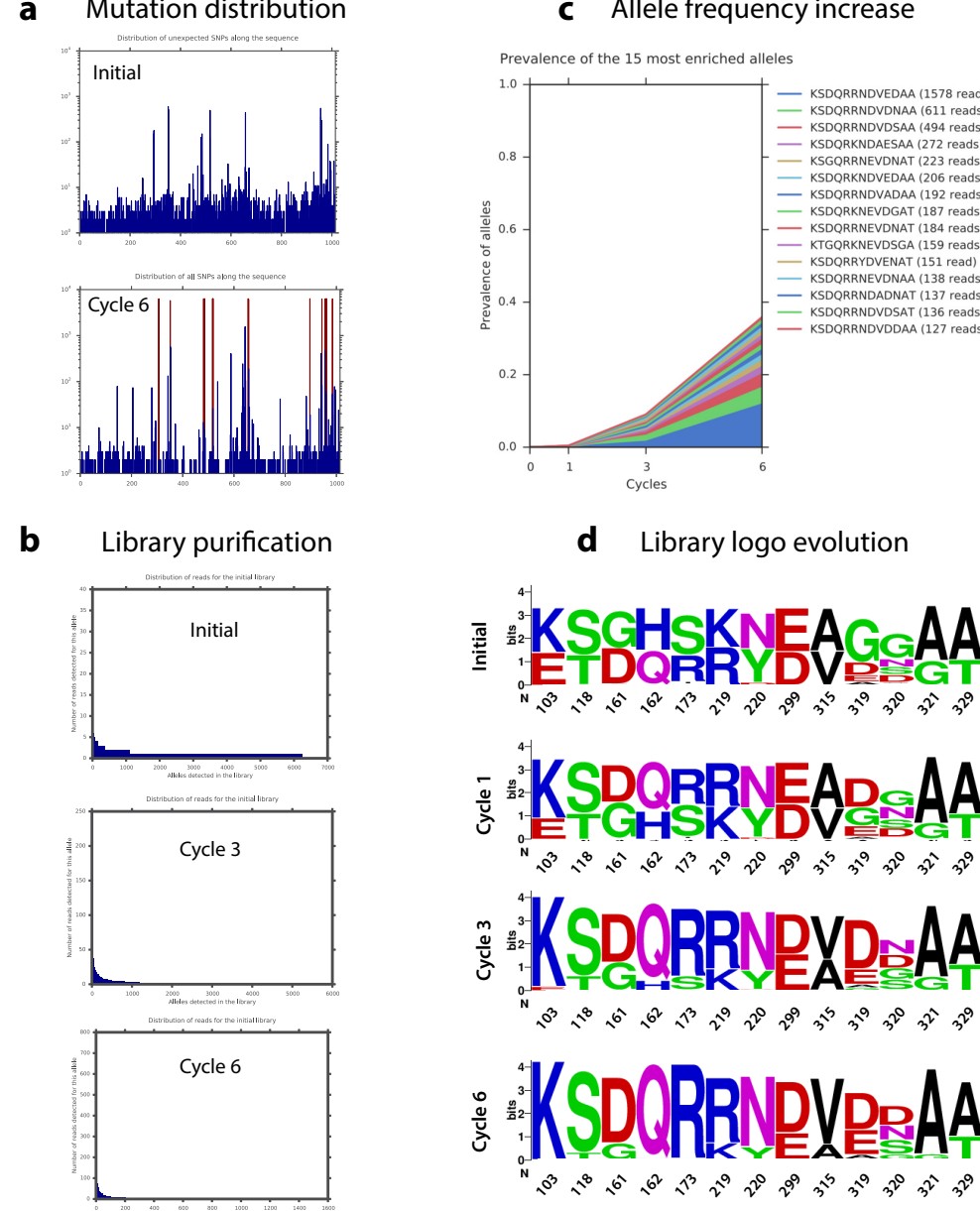

**Appendix 1—figure 7.** Haplotype dynamics during library cycles. (**a**) Distribution of mutations along the *intl1* sequence of all alleles in the final cycle. (**b**) Representation of the number of different alleles found at the initial time point and at cycles 3 and 6. The decrease in allele variety is proof of selection. (**c**) Frequency along the cycles of the 15 most frequent alleles at cycle 6. (**d**) Evolution of the residues at each position of interest during the experiment.

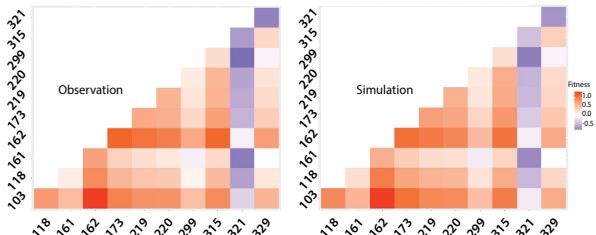

**Appendix 1—figure 8.** Difference in fitness of double mutants compared to the wild type.

