## [Decision Letter]

**Acceptance summary:**

This work examines the mutational basis of an integrase selected for an ancestral activity. The work is carefully executed and demonstrates that it is possible for a 'modern' (insertion-containing) integrase to re-acquire ancestral functionality.

**Decision letter after peer review:**

Thank you for submitting your article "Primary and promiscuous functions coexist during evolutionary innovation through whole protein domain acquisitions" for consideration by *eLife*. Your article has been reviewed by two peer reviewers, and the evaluation has been overseen by a Reviewing Editor and Diethard Tautz as the Senior Editor. The following individual involved in review of your submission has agreed to reveal their identity: Timothy Cooper (Reviewer #3).

The reviewers have discussed the reviews with one another and the Reviewing Editor has drafted this decision to help you prepare a revised submission.

Summary:

Escudero et al. describe evolution of a single strand integron integrase in reverse direction towards its presumed ancestral double strand specificity. The work contains some interesting methodological innovations, such as a method to purge the confounding effects of epistasis from the outcome. Using experimental evolution to recapitulate historical events is a powerful approach. The stated evolutionary implications are that new functions can arise without compromising ancestral functions, and that large insertions can be part of neofunctionalization. This should be of interested to biochemists and evolutionary biologists, although similar conclusions have been reached in several previous studies from other labs

In general terms all reviewers are satisfied that the paper is experimentally sound and addresses a topic of importance and of general interest. However concerns exist, particularly over interpretation of the data, which falls short of supporting the more substantive claims of the manuscript. The data do not show that the insertion contributed to a new function, nor does the data decisively show that the insertion occurred on the background of the ancestral function (rather than occurring after the derived function was acquired by other means). Additionally, the experiments do not show that the historical trajectory was “smooth”. Only that a somewhat smooth trajectory can be obtained in the reverse direction under conditions of strong laboratory selection. The major issues are listed below.

Essential revisions:

1) The Introduction is very difficult to follow, out of kilter with the findings, overlooks relevant work, and makes statements about the evolution of innovation that are not warranted. For example, it is stated that proteins that evolve a new function that replaces the old function have to go through an adaptive valley. This is certainly not true in general. Gene duplications can free one of the duplicates to evolve new functions whilst completely losing the ancestral function. Even if no duplications happen, new functions can evolve without preservation of the ancestral function, as long as the fitness gain of the new function outweighs the loss of the ancestral one. In general, this section gives the impression that we have no worked-out examples of the evolution of novelty. This is not the case. Examples include new protein-protein interactions (Anderson et al., *eLife*, 2016, Baker et al., 2012), the evolution of new substrate specificities (Voordeckers et al., PLOS Biology, 2012), and de novo evolution of enzyme catalysis (Clifton et al., 2018, Kaltenbach et al., 2018), spontaneous domain fusions (Farr et al., 2017). Many of these innovations involved discrete transitions to new functions, without promiscuous intermediates. Nonetheless, we agree that the recombinases described in this work are an impressive example of innovation, and that they are a great model system.

2) The authors imply that the 20 amino acid insertion is causally responsible for the new function, and that this study aims to understand how such a drastic change could contribute to innovation. No evidence seems to exist to support this claim and this needs to be rectified. The present study establishes that the insertion is compatible with the ancestral function, not that it was causal or necessary for the new one. Further, an active site insertion involved in neo-functionalization has been described in detail at least once before (Boucher et al., 2014).

3) Use of an experimental evolution strategy to examine historical transitions is interesting, but it is not clear what is gained by re-evolving ancestral phenotypes from an evolutionary standpoint. Showing that this is possible establishes that the evolution of the new function is reversible in a broad sense, under conditions of strong selection. But specifically the questions the authors pose in their Introduction would seem more adequately addressed by evolving the ancestral function towards the new one. The authors need to elaborate on why the “reverse evolution” approach is superior or equal to other approaches in this context.

4) Results show that it was possible for an integrase to accommodate the I2 domain insertion and keep ancestral activity, but it is necessary to include discussion of the link between this finding and what can really be inferred concerning the ancestral protein (which was presumably quite different from the one relevant at the time of the original insertion), and the constraints relevant at that original time. Discovery that it is possible that the insertion containing the integrase can be selected to a high level of ancestral activity doesn't, by itself, allow a claim that the original insertion didn't compromise that activity. Unclear is how to interpret this potential to the actual changes in activity likely to have been relevant to the evolution of the integrase.

5) Interpretations concerning patterns of epistasis, particularly apparent lack of tendency toward diminishing-returns, need to be re-assessed in light of the experimental findings. In naturally evolving systems, a successful lineage accumulates mutations that have been selected through successive events. As the authors recognise, this is expected to lead to a different pattern of epistasis than for mutations that are assembled without selection (or, even with only purifying selection, which sorts on available combinations, but wasn't designed to facilitate new mutation origination) (e.g., Proc Natl Acad Sci USA. 2015;112: E3226-E3235). Necessary is some discussion of how the (mostly) “assembled” vs. “co-selected” mutation selection regimes might be expected to affect patterns of epistasis relevant to the focal integrase functions. It is surprising not to have seen diminishing-returns, or even more common sign epistasis. What this boils down to is concern over the relevance of the data presented and inferences on the nature of historical evolutionary innovation, which, from the Introduction, is the main motivator for the work. The data here do not convincingly show that the selected path was “smooth” (interpreted as meaning with monotonically increasing activity of the selected function), let alone that this was the case for the historical selection of the ss substrate.

[Editors' note: further revisions were suggested prior to acceptance, as described below.]

Thank you for resubmitting your work entitled "Primary and promiscuous functions coexist during evolutionary innovation through whole protein domain acquisitions" for further consideration by *eLife*. Your revised article has been evaluated by Diethard Tautz (Senior Editor) and a Reviewing Editor.

The manuscript has been improved but there are some remaining issues that need to be addressed before acceptance, as outlined below:

All reviewers are satisfied with the experimental component of the study and agree that the work is interesting and appropriate for *eLife*. However, there remain significant concerns over how the work is motivated and presented, particularly regarding the evolutionary framework.

Part of the problem seems to stem from presenting the work in in a framework that just doesn't fit. Maybe it’s worth taking a step back and starting with what you have done and what it shows, without the higher level context. In essence, what you have done is used a directed evolution strategy based on a powerful screen for enzyme function to generate variants of the extant sequence that have ancestral activity. These mutants and the mutations they harbour were then subject to further analysis to look at interactions among individual mutations. Therein lie some interesting discoveries and possibility to draw certain inferences on the ancestral state. But as the reviewers below point out you state that the ancestral state lacks the I2 α helix, so you really haven't recreated what you believe to be the ancestral sequence.

Additionally, the directed evolution approach can certainly reveal functional states, but as to whether such states were accessible in the context of organism fitness is another matter.

I encourage you to take a look at Dean and Thornton 2007 Nature Genetics, or more recently Hochberg and Thornton 2017 Annu Rev Biophysics, two reviews that place directed evolution experiments in context of what they can and can't say. Both reviews also deal with ancestral reconstruction experiments.

I include below the full set of comments from the reviewers in the hope that these provide further assistance.*Reviewer 1:*

The authors have made changes that generally address my previous comments. Two are outstanding.

I still find it hard to follow the overall presentation of the Introduction. Several points, some discussed at length, seem to have limited relevance to the work presented. For example, the constraint placed by a need to continually perform an original function on evolution of a new function, which I don't think this work is well placed to address. Other points seem more relevant but are discussed briefly. For example, it would be helpful to unpack the contrast between the “new view” of protein genotype-function mapping that allows point mutations to create new functionality without compromising original functionality, with the difficulty in applying this view to “large” mutations.

My main reservation is the overall interpretation that innovation follows (at least, can follow?) a “new model” whereby ancestral function is retained for the majority of the evolutionary steps along the way to a new function. The authors point to data summarized in Figure 7 to support this view. To me, this figure is clearly more consistent with the presented “old view” that the original function quickly declines as a new function evolves. I'm sure that a curve fit with an old view (decreasing concave up) function would fit the data a lot better than a curve fit with a new view (decreasing concave down) function. That said, one characterized allele – Seq7 – is clearly an outlier to the old view, retaining the original function while having a high level of the new function. Does this support that the new view is possible? To me, the answer is far from conclusive – the Seq7 allele has eight amino acid changes relative to the original allele, so that it is not clear it falls along a selectable path connecting original and new integrases-in other words, it is not clear it is a reachable, let alone actual, functional intermediate. I know the authors realize this, but, without that knowledge, I don't see how the observation usefully informs the focal question concerning the pattern of functional changes as a new innovation evolves.*Reviewer 2:*

I think the Introduction and Discussion still contain very unusual renditions of foundational evolutionary concepts that I find hard to agree with. The authors for example describe a “dated” view, in which all adaptive peaks are narrow because proteins are specialized. I think this view, if it ever has been main-stream, has been thoroughly out fashion for over a decade. Neutral networks are known to connect different protein functions. I would point the authors to any of a number of Andreas Wagner's papers. In general, I get the sense of a somewhat false dichotomy between the “new” and whatever is referred to as the “old” view (which very few people hold or have held). Figure 1 A I think is not very helpful in explaining what the authors mean, because it misses the fitness dimension (unless the authors are directly equating activity with fitness, which would be a mistake in many cases).

In their rebuttal and their Discussion, the authors argue that ancestral sequence reconstruction would not be practical here, and so the reverse evolution approach is the most sensible. While this seems to contradict the statement earlier in the paper that the innovation in question happened recently, I did not mean to imply ASR should be used in this case. I meant to ask why the reverse evolution approach is better than evolving the ancestral function (in an extant enzyme!) towards the derived one. This question is important, because it is still not clear to me, how experimental evolution in the reverse direction proves anything about historical evolution the forward direction (especially about smoothness). I am open to the idea that it does, but the arguments levelled in the paper unfortunately did not clarify it for me.

I am also not sure I understand the parsimony arguments about how the insertion must be causal. Parsimony can only assign states to internal nodes (and correspondingly state changes to branches). It is unclear to me how parsimony could ever be used to select among a set of changes along some particular branch the subset of changes that caused a functional change. I think what the authors mean is that the insertion is the most biochemically plausible change to have caused the shift in function. That unfortunately does not seem a very strong argument in the absence of a decisive experiment to me.

I want to re-iterate that the experimental is good, I simply take issue with some of the high-level motivation and interpretation of the results.

---

## [Author Response]

Essential revisions:1) The Introduction is very difficult to follow, out of kilter with the findings, overlooks relevant work, and makes statements about the evolution of innovation that are not warranted. For example, it is stated that proteins that evolve a new function that replaces the old function have to go through an adaptive valley. This is certainly not true in general. Gene duplications can free one of the duplicates to evolve new functions whilst completely losing the ancestral function. Even if no duplications happen, new functions can evolve without preservation of the ancestral function, as long as the fitness gain of the new function outweighs the loss of the ancestral one. In general, this section gives the impression that we have no worked-out examples of the evolution of novelty. This is not the case. Examples include new protein-protein interactions (Anderson et al. eLife, 2016, Baker et al., 2012), the evolution of new substrate specificities (Voordeckers et al., PLOS Biology, 2012), and de novo evolution of enzyme catalysis (Clifton et al., 2018, Kaltenbach et al., 2018), spontaneous domain fusions (Farr et al., 2017). Many of these innovations involved discrete transitions to new functions, without promiscuous intermediates. Nonetheless, we agree that the recombinases described in this work are an impressive example of innovation, and that they are a great model system.

The Introduction has been extensively rewritten and two panels have been added in Figure 1 to make it easier to follow and to find a better balance with the results and the knowledge in the field. Previous works have been acknowledged. The role of gene duplications in fostering innovation has also been mentioned in the Introduction and in the Discussion, which has also been extensively modified.

2) The authors imply that the 20 amino acid insertion is causally responsible for the new function, and that this study aims to understand how such a drastic change could contribute to innovation. No evidence seems to exist to support this claim and this needs to be rectified. The present study establishes that the insertion is compatible with the ancestral function, not that it was causal or necessary for the new one. Further, an active site insertion involved in neo-functionalization has been described in detail at least once before (Boucher et al., 2014).

There is indeed no direct evidence of causality, because a deletion of the I2 domain abolishes all activity of IntI1, but there is a wealth of indirect data strongly supporting this assumption as the most parsimonious one. We have discussed this in the first paragraph of the Discussion.

3) Use of an experimental evolution strategy to examine historical transitions is interesting, but it is not clear what is gained by re-evolving ancestral phenotypes from an evolutionary standpoint. Showing that this is possible establishes that the evolution of the new function is reversible in a broad sense, under conditions of strong selection. But specifically the questions the authors pose in their Introduction would seem more adequately addressed by evolving the ancestral function towards the new one. The authors need to elaborate on why the “reverse evolution” approach is superior or equal to other approaches in this context.

We have added a section called “Limitations and other considerations” in which we argue that the approach of resurrecting the ancestral sequence is highly questionable in this case, because of long evolutionary time scales, the small number of IntI sequences available, and the lack of intermediates. Nevertheless, we do acknowledge the benefits of such strategy and the limitations in our work that derive from the lack of such an ancestor.

4) Results show that it was possible for an integrase to accommodate the I2 domain insertion and keep ancestral activity, but it is necessary to include discussion of the link between this finding and what can really be inferred concerning the ancestral protein (which was presumably quite different from the one relevant at the time of the original insertion), and the constraints relevant at that original time.

We have added a section in the Discussion entitled “Possible scenario for the emergence of integron integrases” in which we speculate on how the acquisition of an I2 domain could have taken place in IntI ancestors, their potential function and the constraints they might have found and the possible solutions to I2-acquisition deleterious effects.

Discovery that it is possible that the insertion containing the integrase can be selected to a high level of ancestral activity doesn't, by itself, allow a claim that the original insertion didn't compromise that activity.

We acknowledge this limitation of our work. We have also added a new Figure 7 in which we show that two mutants (Seq5 and Seq7) have very high and similar levels of ancestral function but relevant differences in modern activity. This suggests that after I2 acquisition, the ancestor could have gained in modern activity without impacting the ancestral activity, to a point where the modern activity is enough to be selected for.

Unclear is how to interpret this potential to the actual changes in activity likely to have been relevant to the evolution of the integrase.

As mentioned before, we discuss a possible scenario for the acquisition of I2 in the ancestor. It is known that Xer recombinases (that undo chromosome dimers) are the closest relatives to IntIs. In the plausible scenario of IntI ancestor having a similar activity to Xer proteins, being capable of acquiring I2 without completely disrupting its main activity might have been key to buy the time for the neofunctionalization process to take place. We also envisage that I2 could initially serve to tamper the deleterious effect of the recombination of IMEX elements, potentially compensating for any negative impact on IntIs ancestor main activity.

5) Interpretations concerning patterns of epistasis, particularly apparent lack of tendency toward diminishing-returns, need to be re-assessed in light of the experimental findings. In naturally evolving systems, a successful lineage accumulates mutations that have been selected through successive events. As the authors recognise, this is expected to lead to a different pattern of epistasis than for mutations that are assembled without selection (or, even with only purifying selection, which sorts on available combinations, but wasn't designed to facilitate new mutation origination) (e.g., Proc Natl Acad Sci USA. 2015;112: E3226-E3235). Necessary is some discussion of how the (mostly) “assembled” vs. “co-selected” mutation selection regimes might be expected to affect patterns of epistasis relevant to the focal integrase functions. It is surprising not to have seen diminishing-returns, or even more common sign epistasis. What this boils down to is concern over the relevance of the data presented and inferences on the nature of historical evolutionary innovation, which, from the Introduction, is the main motivator for the work.

We address this in the text where we provide plausible explanations for the lack of diminishing returns: (i) that selection is acting on different activities, (ii) that our measure of fitness is averaged over all genetic backgrounds in the library, or (iii) that it is consequence of starting very far from optimality. We also acknowledge the different expectations on the prevalence of epistasis in the “far from optimal” scenario for both co-selected or combined mutations. We argue that when mutations arise in a far from optimal background, epistatic interactions can be barely detectable, and hence the pattern of diminishing returns might be absent at this level and become apparent when the protein gets closer to optimality, -where epistatic patterns become more prevalent and complex.

The data here do not convincingly show that the selected path was “smooth” (interpreted as meaning with monotonically increasing activity of the selected function), let alone that this was the case for the historical selection of the ss substrate.

We have added Figure 7 in the Discussion in which we show that some of our mutants do follow a path in which the asymmetry of trade-offs is intense enough to be compatible with a smooth (interpreted as “through functional intermediates”) initial transition between functions. This is not the case for all mutants, but we have to bear in mind that there has been absolutely no pressure to keep the modern activity in our experiments.

[Editors' note: further revisions were suggested prior to acceptance, as described below.]

Reviewer 1:The authors have made changes that generally address my previous comments. Two are outstanding.I still find it hard to follow the overall presentation of the Introduction. Several points, some discussed at length, seem to have limited relevance to the work presented. For example, the constraint placed by a need to continually perform an original function on evolution of a new function, which I don't think this work is well placed to address. Other points seem more relevant but are discussed briefly. For example, it would be helpful to unpack the contrast between the “new view” of protein genotype-function mapping that allows point mutations to create new functionality without compromising original functionality, with the difficulty in applying this view to “large” mutations.

As mentioned previously, we have extensively changed the Introduction, and taken out the part about the old/new view.

My main reservation is the overall interpretation that innovation follows (at least, can follow?) a “new model” whereby ancestral function is retained for the majority of the evolutionary steps along the way to a new function. The authors point to data summarized in Figure 7 to support this view. To me, this figure is clearly more consistent with the presented “old view” that the original function quickly declines as a new function evolves. I'm sure that a curve fit with an old view (decreasing concave up) function would fit the data a lot better than a curve fit with a new view (decreasing concave down) function. That said, one characterized allele – Seq7 – is clearly an outlier to the old view, retaining the original function while having a high level of the new function. Does this support that the new view is possible? To me, the answer is far from conclusive – the Seq7 allele has eight amino acid changes relative to the original allele, so that it is not clear it falls along a selectable path connecting original and new integrases-in other words, it is not clear it is a reachable, let alone actual, functional intermediate. I know the authors realize this, but, without that knowledge, I don't see how the observation usefully informs the focal question concerning the pattern of functional changes as a new innovation evolves.

We agree with reviewer 1 that Figure 7 (especially the curve fitting the points) seems to be an effort to convey the message that innovation through functional intermediates is the most common pathway, which is unreasonable. This was not the message we were aiming to convey, and have hence erased the curve. Our intention with Figure 7 is to depict that, although most mutations fall within the lower part of the graph, there are some clear outliers on the top part (this is precisely the old versus new model dichotomy that we have taken out from the Introduction).

Focusing specifically on mutants Seq 5 and Seq 7 we observe two results that are relevant.

1) The ancestral activity is not necessarily compromised by the acquisition of I2. The fact that I2-containing proteins can have high levels of ancestral activity supports the possibility of a route through functional intermediates.

2) Seq 5 has the highest levels of ancestral activity and the lowest levels of modern one (it actually shows an almost symmetric trade-off between functions). This mutant is the closest representative in our experiments to a hypothetical “early innovative state” of an integrase that would have recently acquired the I2 domain. It is of note that Seq5 already has detectable modern activity (in our very sensitive assays), but we do not assume that this would have been necessarily the case initially. Comparing Seq5 to Seq7, mutants that are only 3 mutations away (not eight), shows that an integrase in an early innovative state can increase its modern activity 100x without a trade off in ancestral activity, supporting the general message that coexistence of functions (promiscuity) is also possible in proteins suffering large rearrangements ant that it fosters innovation.

It is of note that we do not aim to impose the view that any of these pathways was the real one, but rather that they are possible. We are conscious of the speculative nature of this section and that is the reason why Figure 7 is only mentioned in the Discussion section.

Reviewer 2:I think the Introduction and Discussion still contain very unusual renditions of foundational evolutionary concepts that I find hard to agree with. The authors for example describe a “dated” view, in which all adaptive peaks are narrow because proteins are specialized. I think this view, if it ever has been main-stream, has been thoroughly out fashion for over a decade. Neutral networks are known to connect different protein functions. I would point the authors to any of a number of Andreas Wagner's papers. In general, I get the sense of a somewhat false dichotomy between the “new” and whatever is referred to as the “old” view (which very few people hold or have held). Figure 1 A I think is not very helpful in explaining what the authors mean, because it misses the fitness dimension (unless the authors are directly equating activity with fitness, which would be a mistake in many cases).

We have extensively modified the Introduction and erased the old/new view dichotomy.

In their rebuttal and their Discussion, the authors argue that ancestral sequence reconstruction would not be practical here, and so the reverse evolution approach is the most sensible. While this seems to contradict the statement earlier in the paper that the innovation in question happened recently, I did not mean to imply ASR should be used in this case. I meant to ask why the reverse evolution approach is better than evolving the ancestral function (in an extant enzyme!) towards the derived one. This question is important, because it is still not clear to me, how experimental evolution in the reverse direction proves anything about historical evolution the forward direction (especially about smoothness). I am open to the idea that it does, but the arguments levelled in the paper unfortunately did not clarify it for me.

As we mentioned briefly in the Discussion (a new sentence has now been added to include also extant enzymes), we believe that evolving de novo the modern ss-recombination function in an extant or resurrected enzyme specialized in the ancestral ds-recombination activity is beyond reach in the laboratory. Spontaneous insertions of long domains are rare enough in the lab to make this approach extremely risky, especially if they have to bring a novel function, and not simply change the protein’s conformation locally. Additionally, designing the whole experiment to detect unequivocally single strand DNA recombination would be extremely complex. Hypothetically, one would use as a starting point, Xer recombinases (the closest relatives to integrases), which in most species -and certainly the ones we use as models in the lab- act as heterodimers of XerC and XerD. These enzymes recognize a canonical ds-DNA site called *dif*. We would hence need to modify the *dif* site to try to convert it into an *attC*-like site, including the EHBs. We would need to develop a selection method to detect single strand recombination (avoiding any noise with ds-recombination) on this site, to use as selective pressure in the lab to obtain insertions in Xer conferring ss-recombination capabilities. This, assuming our new *dif* site will behave as we expect, which we cannot test previously. If we managed to retrieve such events, we would then need to verify that mutations found in any of the Xer proteins really lead to the recognition of the new *dif* site as a single strand, and specifically for the EHBs. This is crucial because mutated Xer proteins (or WT, for that matter) could simply recognize the folded site as a double strand, which is what happens with the CTX phage in *Vibrio cholerae*, and all our setup would be in vain. To do so we would need to obtain the structure of the mutant bound to the site through crystallography, as was done for *V. cholerae’s* integron integrase (but has never been obtained for any other integrase despite years of effort). Even then, one might be prudent enough to suggest that mutations are not necessarily causal to ss-DNA recognition, so we would have to prove that the ancestral Xer protein does not bind the ss-dif sites in the same way. This would be really hard, because it is even more unlikely that we would obtain such structure. So altogether it seems that the only viable approach to study the phenomenon is the way we have done it, given the luck of having both activities in an extant enzyme.

I am also not sure I understand the parsimony arguments about how the insertion must be causal. Parsimony can only assign states to internal nodes (and correspondingly state changes to branches). It is unclear to me how parsimony could ever be used to select among a set of changes along some particular branch the subset of changes that caused a functional change. I think what the authors mean is that the insertion is the most biochemically plausible change to have caused the shift in function. That unfortunately does not seem a very strong argument in the absence of a decisive experiment to me.

Our use of parsimony humbly refers to the simplest explanation. Our rationale is that if there is a domain that does not exist in any other recombinase; no other recombinase does ss/ds-DNA recombination, and the crystal structure shows that the domain is responsible for ss-recognition in integrases. We hence think that the simplest explanation is that the domain brought the function. One can argue that it didn’t and that the function came later, but that implies that the region where the I2 was initially inserted is flexible enough to accommodate function-less insertions without being deleterious for the protein function. This does not seem to be the case, because there are no such insertions in the same region of 105 members of the Y-recombinase family (Nunes-Duby et al., 1998). Again, proving this experimentally is unfortunately beyond our current reach, so we hope to have clearly stated that these are assumptions and to have toned them down enough to make them reasonable.